# Entropy-based metrics for predicting choice behavior based on local response to reward

Ethan Trepka[1,6], Mehran Spitmaan [1,6], Bilal A. Bari[2,3,4], Vincent D. Costa [5], Jeremiah Y. Cohen [2,3,4] & Alireza Soltani [1✉]

For decades, behavioral scientists have used the matching law to quantify how animals distribute their choices between multiple options in response to reinforcement they receive. More recently, many reinforcement learning (RL) models have been developed to explain choice by integrating reward feedback over time. Despite reasonable success of RL models in capturing choice on a trial-by-trial basis, these models cannot capture variability in matching behavior. To address this, we developed metrics based on information theory and applied them to choice data from dynamic learning tasks in mice and monkeys. We found that a single entropy-based metric can explain 50% and 41% of variance in matching in mice and monkeys, respectively. We then used limitations of existing RL models in capturing entropy-based metrics to construct more accurate models of choice. Together, our entropy-based metrics provide a model-free tool to predict adaptive choice behavior and reveal underlying neural mechanisms.

[1] Department of Psychological and Brain Sciences, Dartmouth College, Hanover, NH, USA. [2] The Solomon H. Snyder Department of Neuroscience, The Johns Hopkins University School of Medicine, Baltimore, MD, USA. [3] Brain Science Institute, The Johns Hopkins University School of Medicine, Baltimore, MD, USA. [4] Kavli Neuroscience Discovery Institute, The Johns Hopkins University School of Medicine, Baltimore, MD, USA. [5] Department of Behavioral Neuroscience, Oregon Health and Science University, Portland, OR, USA. [6]These authors contributed equally: Ethan Trepka, Mehran Spitmaan. ✉email: alireza.soltani@dartmouth.edu

How do we distribute our time and choices between the many options or actions available to us? Around 60 years ago, Richard Herrnstein strived to answer this question based on one of the key ideas of behaviorism; that is, the history of reinforcement is the most important determinant of behavior. He proposed a simple rule called the matching law stating that the proportion of time or responses that an animal allocates to an option or action matches the proportion of reinforcement they receive from those options or actions[1]. The matching law has been shown to explain global choice behavior across many species[2] including pigeons[3–6], mice[7–9], rats[10–12], monkeys[13–18], and humans[19–24], in a wide range of tasks, including concurrent variable interval, concurrent variable ratio, probabilistic reversal learning, and so forth. A common finding in most studies, however, has been that animals undermatch, corresponding to selection of the better stimulus or action less than it is prescribed by the matching law. Such deviation from matching often corresponds to suboptimal behavior in terms of total harvested rewards, pointing to adaptive mechanisms beyond reward maximization.

The matching law is a global rule but ultimately should emerge from an interaction between choice and learning strategies, resulting in local (in time) adjustments of choice behavior to reinforcement obtained in each trial. Accordingly, many studies have tried to explain matching based on different learning mechanisms. Reinforcement-learning (RL) models are particularly useful because they can simulate changes in behavior due to reward feedback. Consequently, many studies on matching have focused on developing RL models that can generate global matching behavior based on local learning rules[14,15,25–32]. These RL models are often augmented with some components in addition to stimulus- or action-value functions to improve fit of choice behavior on a trial-by-trial basis. For example, the models could include learning the reward-independent rate of choosing each option[15], adopting win-stay lose-switch (WSLS) policies[27,28], or learning on multiple timescales[31]. Although these models all provide compelling explanations of the emergence of matching behavior, it remains unclear how they compare in terms of fitting local choice behavior and the extent to which they replicate observed variability in matching behavior. This could result in misinterpretation or missing important neural mechanisms underlying matching behavior in particular and adaptive behavior more generally[33,34]. Therefore, after decades of research on matching behavior, it is still not fully understood how such a fundamental law of behavior emerges as a result of local response to reward feedback.

In this work, we propose a set of metrics based on information theory that can summarize trial-by-trial response to reward feedback and predict global matching behavior. To test the utility of our metrics, we apply them to large sets of behavioral data in mice and monkeys during two very different dynamic learning tasks. We find that in both mice and monkeys, our entropy-based metrics can predict deviation from matching better than existing measures. Specifically, we find the strongest link between undermatching and the consistency of choice strategy (stay or switch) in response to receiving no reward after selection of the worse option in both species. Finally, we use shortcomings of purely RL models in capturing the pattern of entropy-based metrics in our data to construct multicomponent models that integrate reward- and option-dependent strategies with standard RL models. We show that these models can capture both trial-by-trial choice data and global choice behavior better than the existing models, thus revealing additional mechanisms involved in adaptive learning and decision making.

## Results
### Mice and monkeys dynamically adjust their behavior to changes in reward probabilities.
To study learning and decision making in dynamic reward environments, we examined choice behavior of mice and monkeys during two different probabilistic reversal learning tasks. Mice selected between two actions (licking left and right) that provided reward with different probabilities, and these probabilities changed between blocks of trials without any signal to the animals[9] (Fig. 1a; see Methods for more details). Block lengths were drawn from a uniform distribution that spanned a range of 40 to 80 trials. Here, we focused on the majority (469 out of 528) of sessions in which two sets of reward probabilities (equal to 0.4 and 0.1, and 0.4 and 0.05) were used. We refer to these reward schedules as 40/10 and 40/5 reward schedules (1786 and 1533 blocks with 40/5 and 40/10 reward schedules, respectively). Rewards were baited such that if reward was assigned on a given side and that side was not selected, reward would remain on that side until the next time that side was selected. Due to baiting, the probability of obtaining reward on the unchosen side increased over time as during foraging in a natural environment. As a result, selecting the worse side (side with lower base reward rate) occasionally can improve the overall total harvested reward. In total, 16 mice performed 469 sessions of the two-probability version of the task for a total of 3319 blocks and 189,199 trials.

In a different experiment, monkeys selected between pairs of stimuli (a circle or square with variable colors) via saccades and received a juice reward probabilistically[35] (Fig. 1b; see Methods for more details). In superblocks of 80 trials, the reward probabilities assigned to each stimulus reversed randomly between trials 30 and 50, such that the more-rewarding stimulus became the less-rewarding stimulus. We refer to trials before and after a reversal as a block. Monkeys completed multiple superblocks per each session of the experiment wherein the reward probabilities assigned to the better and worse stimuli were equal to 0.8 and 0.2, 0.7 and 0.3, or 0.6 and 0.4, which we refer to as 80/20, 70/30, and 60/40 reward schedules. In contrast to the task used in mice, rewards were not baited. Here, we only analyze data from the 80/20 and 70/30 reward schedules as they provide two levels of reward uncertainty similar to the experiment in mice. In total, 4 monkeys performed 2212 blocks of the task with the 80/20 and 70/30 reward schedules for a total of 88,480 trials.

We found that in response to block switches, both mice and monkeys rapidly adjusted their choice behavior to select the better option (better side or stimulus in mice and monkeys, respectively) more often (Fig. 1c, d). However, the fraction of times they chose the better option fell below predictions made by the matching law, even at the end of the blocks (Fig. 1e, f). More specifically, the relative selection of the better option (i.e., choice fraction) was often lower than the ratio of reward harvested on the better option to the overall reward harvested (i.e., reward fraction), corresponding to undermatching behavior. Therefore, we next explored how undermatching depends on choice- and reward-dependent strategies.

### Mice and monkeys exhibit highly variable undermatching behavior.
To better examine matching behavior, we used the difference between the relative choice and reward fractions for each block of trials to define "deviation from matching" (see Eqs. 1, 2 in Methods; Fig. 2a, d). Based on our definition, negative and positive values for deviation from matching correspond to undermatching and overmatching, respectively. Undermatching occurs when the relative choice fraction is smaller than the relative reward fraction for reward fractions larger than 0.5, or the relative choice fraction is larger than the relative reward fraction when the latter is smaller than 0.5. Overmatching occurs when the relative choice fraction is larger than the relative reward fraction for reward fractions larger than 0.5, or the relative choice

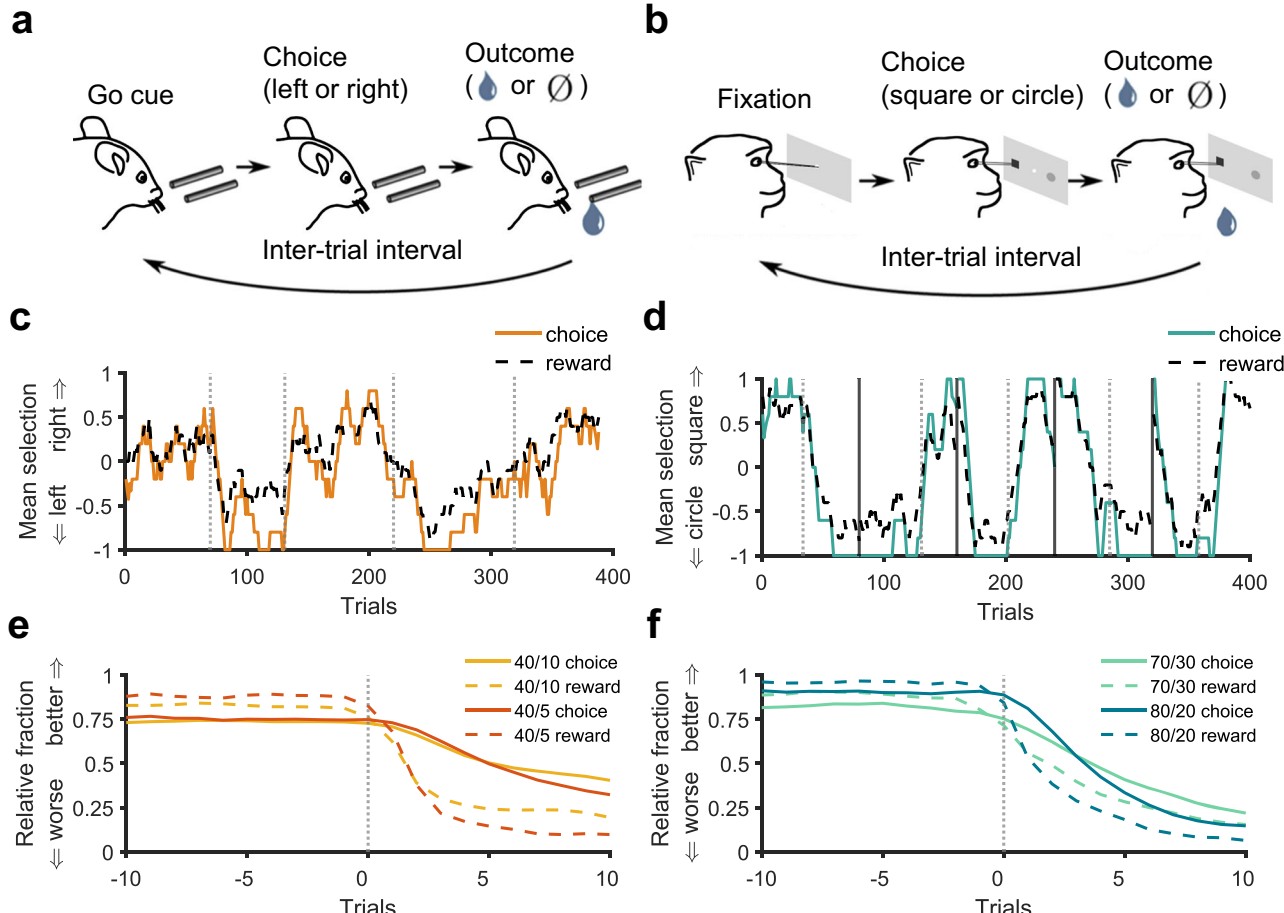

**Fig. 1 Schematic of the experimental paradigms in mice and monkeys and basic behavioral results. a**, **b** Timeline of a single trial during experiments in mice (**a**) and monkeys (**b**). To initiate a trial, mice received an olfactory go cue (or no-go cue in 5% of trials) (**a**), and monkeys fixated on a central point (**b**). Next, animals chose (via licks for mice and saccades for monkeys) between two options (left or right tubes for mice and circle or square for monkeys) and then received a reward (drop of water and juice for mice and monkeys, respectively) probabilistically based on their choice. **c**, **d** Average choice and reward using a sliding window with a length of 10 for a representative session in mice (**c**) and five superblocks of a representative session in monkeys (**d**). Mean selection of 1 and −1 correspond to 100% selection of or 100% reward on the right and left in mice (square and circle stimuli in monkeys), respectively, and mean selection of 0 corresponds to equal selection or reward on the two choice options. Vertical gray dashed lines indicate trials where reward probabilities reversed. Vertical gray solid lines indicate divisions between superblocks in the monkey experiment. **e**, **f** Average relative choice and reward fractions around block switches using a non-causal smoothing kernel with a length of three separately for all blocks with a given reward schedule in mice (**e**) and monkeys (**f**). The better (or worse) option is the better (or worse) option prior to the block switch. Trial zero is the first trial with the reversed reward probabilities. Average choice fractions for the better option (better side or stimulus) are lower than average reward fractions for that option throughout the block for both mice and monkeys, corresponding to undermatching behavior.

fraction is smaller than the relative reward fraction when the latter is smaller than 0.5. Undermatching could happen because the animal does not detect the more-rewarding stimulus or action, poor credit assignment, or due to too much stochasticity in choice. In contrast, overmatching is characterized by selecting the better option more frequently than is prescribed based on perfect matching. In the task used in monkeys, overmatching was not possible by design (except due to random fluctuations in reward assignment) and harvested rewards could be maximized by selecting the better stimulus all the time, corresponding to matching. In contrast, overmatching was possible in the reversal learning tasks with baited rewards (e.g., task used in mice).

Consistent with previous studies on matching behavior, we found significant undermatching in mice in both the 40/10 and 40/5 reward schedules (Wilcoxon signed-rank test; $40/10: Z = -31.2, p = 1.53 \times 10^{-213}; 40/5: Z = -35.0, p = 8.75 \times 10^{-269};$ Fig. 2b, c). Similarly, we found significant undermatching in monkeys in both the 70/30 and 80/20 reward schedules

(Wilcoxon signed-rank test; $70/30: Z = -27.02, p = 9.74 \times 10^{-161}; 80/20: Z = -27.06, p = 3.06 \times 10^{-161};$ Fig. 2e, f). In addition, average undermatching for mice was significantly larger in the 40/5 reward schedule than the 40/10 reward schedule, whereas average undermatching for monkeys was not significantly different in the 70/30 and 80/20 schedules (two-sided independent $t$ test; Mice: $p = 7.93 \times 10^{-36}, d = 0.44$; Monkeys: $p = 0.19, d = 0.06$; Supplementary Fig. 1f). More importantly, undermatching was highly variable in both reward schedules for both mice and monkeys (Fig. 2b, c, e, f). To understand the nature of this variability, we examined whether existing behavioral metrics and RL models can predict the observed deviation from matching.

**Existing behavioral metrics only partially explain variability in matching.** To examine the relationship between existing behavioral metrics and undermatching, we first performed stepwise

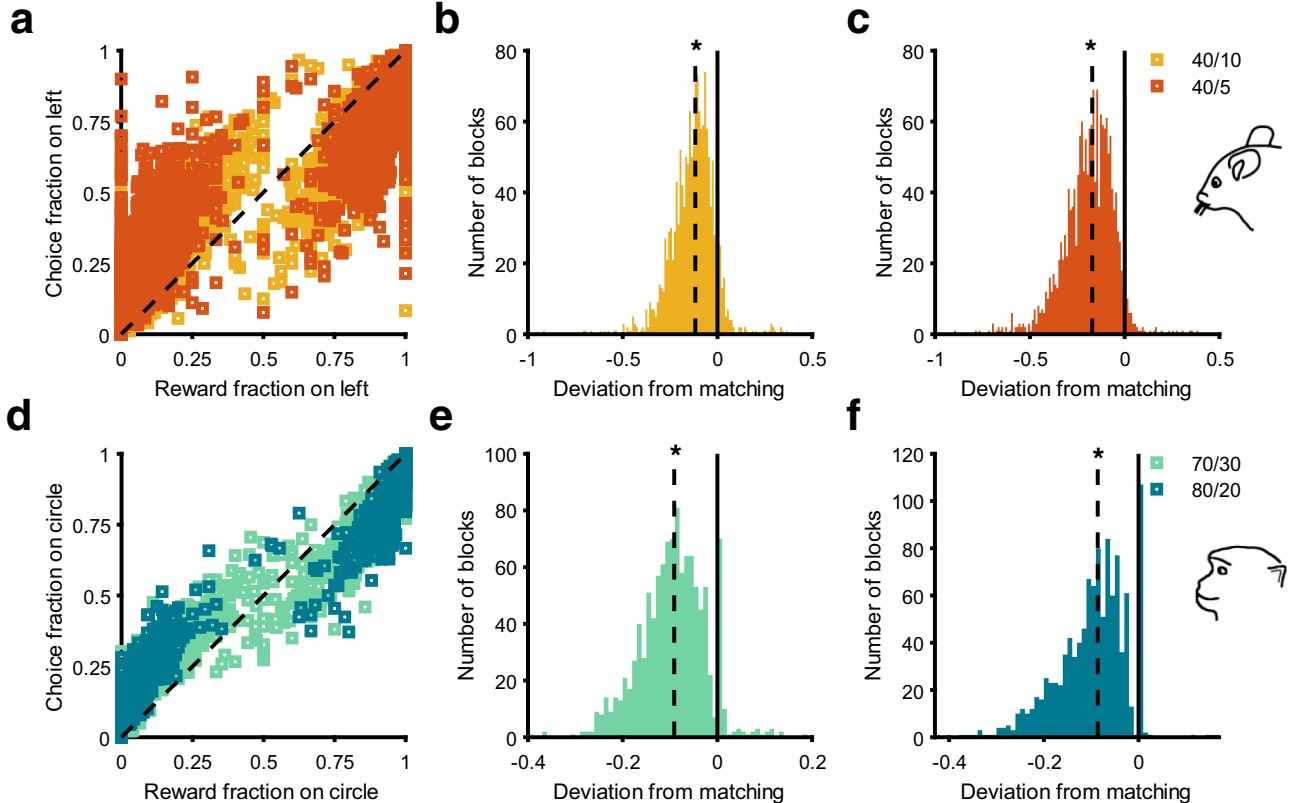

**Fig. 2 Mice and monkeys exhibit highly variable but significant undermatching. a** Plot shows relative reward fraction versus relative choice fraction across all blocks separately for each reward schedule in mice. The black dashed line represents the identity line. For relative reward fractions >0.5, nearly all points remain below the identity line, indicating a relative choice fraction smaller than reward fraction for the better option (undermatching). Similarly, points above the identity line for relative reward fractions <0.5 indicate undermatching. **b, c** Histograms show deviation from matching for the 40/10 (**b**) and 40/5 reward schedules (**c**) in mice. The solid black line indicates 0, corresponding to perfect matching. The dashed black lines are the median deviation from matching. Asterisks indicate significance based on a two-sided Wilcoxon signed-rank test (40/10: $p = 1.53 \times 10^{-213}$; 40/5: $p = 8.75 \times 10^{-269}$). **d–f** Similar to (**a–c**), but for monkeys with 70/30 and 80/20 reward schedules. Asterisks indicate significance based on a two-sided Wilcoxon signed-rank test (70/30: $p = 9.74 \times 10^{-161}$; 80/20: $p = 3.06 \times 10^{-161}$). Because of random fluctuations in local reward probabilities, overmatching occurred in a minority of cases.

multiple regressions to predict deviation from matching for both mice and monkeys based on commonly used behavioral metrics including: $p(win), p(stay), p(stay,|win)$, and $p(switch|lose)$. The threshold for adding a predictor was set at $p < 0.0001$ (see "Methods" for more details and Supplementary Note 1 for regression equations). These regression models explained 31% and 34% of the variance in deviation from matching for mice and monkeys, respectively, which are significant but unsurprising amounts of overall variance (mice: *Adjusted* $R^2 = 0.31$; monkeys: *Adjusted* $R^2 = 0.34$).

We next included the Repetition Index (RI) on the better ($RI_B$) and worse ($RI_W$) options (side or stimulus), which measure the tendency to stay beyond chance on the better and worse options[36] to predict undermatching. To that end, we conducted additional stepwise multiple regressions that predicted deviation from matching using: $RI_B, RI_W, p(win), p(stay), p(stay|win)$, and $p(switch|lose)$ as predictors. These models explained 48% and 49% of the variance in deviation from matching for mice and monkeys, respectively (mice: *Adjusted* $R^2 = 0.48$; *monkeys*: *Adjusted* $R^2 = 0.49$). Thus, including $RI_B$ and $RI_W$ enabled us to account for additional 17% and 15% of variance, suggesting that staying beyond chance on both the better and worse choice options is a significant contributor to undermatching behavior.

Together, these results illustrate that undermatching is correlated with tendency to stay beyond chance (measured by RI) and response

to reward feedback in terms of stay or switch (measured by win-stay and lose-switch). However, win-stay and lose-switch are not strong predictors of undermatching because their relative importance depends on the overall probability of winning. For example, if $p(win)$ is high, lose-switch is less useful for predicting behavior because response to loss represents strategy in a small subset of trials. Although win-stay, lose-switch, $p(win)$, and $p(stay)$ contain all the information necessary to compute the dependence of strategy on reward, this requires interpretation of all four metrics in conjunction and may depend on nonlinear relationships that are challenging to intuit or capture with regression. To overcome these issues, we propose metrics to quantify changes in strategy due to reward outcome using information theory.

**Behavioral metrics based on information theory**. To better capture the dependence of staying (or similarly switching) strategy on reward outcome, we developed a series of model-free behavioral metrics based on Shannon's information entropy[37]. The Shannon information entropy of a random variable $X$ conditioned on $Y$, denoted $H(X|Y)$, captures the surprise or uncertainty of $X$ given knowledge of the values of $Y$. Lower information entropies correspond with decreased uncertainty in the variable under consideration and thus consistency in utilized strategy (see below).

First, we define the entropy of reward-dependent strategy (ERDS) that measures the dependence of adopting a response strategy on reward outcome. Formally, ERDS is the information entropy of response strategy conditioned on reward outcome in the previous trial, $H(str,|rew)$ (see "Methods" for more details). Therefore, ERDS quantifies the amount of information needed to explain an animal's strategy (e.g., choosing the option selected in the previous trial) given knowledge of reward outcome (e.g., whether the animal won or lost in the previous trial). Lower ERDS values indicate more consistent response to reward feedback.

In its simplest formulation for measuring the effect of reward in the previous trial on staying or switching, ERDS is a function of win-stay, lose-switch, and $p(win)$ (see Eq. (8) in "Methods"). As win-stay and lose-switch move further from 0.5, ERDS decreases, reflecting increased consistency of reward-dependent strategy (Fig. 3a). Moreover, $p(win)$ modulates the effects of win-stay and lose-switch on ERDS (Fig. 3b, c). As p(win) decreases, the influence of win-stay on ERDS decreases, reflecting that win-stay is less relevant to overall response to reward feedback when winning rarely occurs. Similarly, as $p(lose) (= 1 - p(win))$ decreases, the influence of lose-switch on ERDS decreases, reflecting that lose-switch is less relevant to response to reward feedback. Because of these properties, ERDS corrects for the limitations of win-stay and lose-switch. Also, as stay (or switch) strategy becomes more independent of reward outcome, ERDS increases because reward outcome provides no information about strategy.

ERDS can be decomposed into $ERDS_+$ and $ERDS_-$ to measure the specific effects of winning and losing in the preceding trial, respectively (Fig. 3b, c; see Methods). More specifically, $ERDS_+$ is the entropy of win-dependent strategy, and $ERDS_-$ is the entropy of loss-dependent strategy. Therefore, comparing $ERDS_+$ and $ERDS_-$ provides information about the relative contributions of win-dependent strategy and loss-dependent strategy to the overall reward-dependent strategy.

In addition to conditioning stay strategy on reward outcome in the preceding trial, we can also condition stay strategy on selection of the better or worse choice option (stimulus or action) in the previous trial. The resulting entropy of option-dependent strategy (EODS), $H(str,|opt)$, captures the dependence of stay (or switch) strategy on the selection of the better or worse option in the previous trial (Fig. 3d). EODS depends on $p(choose\ better)$, $p(stay|choose\ better)$, and $p(switch|choose\ worse)$ and moreover, can be decomposed into $EODS_B$ and $EODS_W$ based on selection of the better or worse option, respectively (Fig. 3e, f).

Finally, to capture the dependence of response strategy on reward outcome and the previously selected option in a single metric, we computed the entropy of reward- and option-dependent strategy (ERODS), $H(str,|,rew,opt)$. ERODS depends on the probabilities of adopting a response strategy conditioned on all combinations of reward outcome and option selected in the previous trial (see "Methods" for more details). ERODS has similar properties to ERDS and EODS and can be interpreted in a similar fashion. Low ERODS values indicate that stay (or switch) strategy consistently depends on combinations of win/lose and the option selected in the previous trial; for example, winning and choosing the better side in the previous trial. ERODS can be decomposed either by the better or worse option ($ERODS_B$ and $ERODS_W$), by win or loss ($ERODS_+$ and $ERODS_-$), or by both ($ERODS_{B+}$, $ERODS_{W+}$, $ERODS_{B-}$, and $ERODS_{W-}$).

To summarize, we propose three metrics, ERDS, EODS, and ERODS that capture the dependence of response strategy on reward outcome and/or selected option in the preceding trial. Each metric can be decomposed into components that provide important information about the dependence of stay (or switch) strategy on winning or losing and/or choosing the better or worse option in the preceding trial (see Supplementary Table 1 for summary). We next show how these entropy-based metrics can predict deviation from matching behavior and further be used to construct more successful RL models.

**Deviation from matching is highly correlated with entropy-based metrics**. To test the relationship between the observed undermatching and our entropy-based metrics, we next computed correlations between all behavioral metrics and deviation from matching (Supplementary Fig. 2). We found that nearly all entropy-based metrics were significantly correlated with deviation from matching. Out of all behavioral metrics tested, deviation from matching showed the strongest correlation with $ERODS_{W-}$ based on both parametric (Pearson; mice: $r = -0.71, p < 10^{-300}$; monkeys: $r = -0.64, p = 10^{-231}$) and non-parametric (Spearman; mice: $r = -0.78, p < 10^{-300}$; monkeys: $r = -0.75, p < 10^{-300}$) tests in both mice and monkeys (Fig. 4). The size of the correlation between $ERODS_{W-}$ and deviation from matching is remarkable because it indicates that a single metric can capture more than 50% and 41% of the variance in deviation from matching in mice and monkeys, respectively. This finding suggests that undermatching occurs when animals lose when selecting the worse option and respond inconsistently to those losses.

Out of all existing behavioral metrics tested, the probability of winning and probability of staying had the two strongest correlations with deviation from matching for mice and monkeys, respectively, but each metric individually only captured about 25% of variance in deviation from matching (Fig. 4). The correlation between the probability of winning (total harvested rewards) and deviation from matching was positive such that increased total harvested rewards corresponded with less undermatching.

In addition to $ERODS_{W-}$, $EODS_W$ was also highly correlated with deviation from matching (Pearson: mice: $r = -0.60, p < 10^{-300}$; monkeys: $r = -0.53, p = 9.35 \times 10^{-150}$; Spearman: mice: $r = -0.67, p < 10^{-300}$, monkeys: $r = -0.67, p = 1.18 \times 10^{-265}$) as was $ERDS_-$ (Pearson: mice: $r = -0.43, p = 6.72 \times 10^{-156}$; monkeys: $r = -0.57, p = 1.94 \times 10^{-191}$; Spearman: mice: $r = -0.47, p = 7.31 \times 10^{-198}$; monkeys: $r = -0.63, p = 1.06 \times 10^{-240}$). Overall, these results show that global deviation from matching was most strongly correlated with the consistency of response after selection of the worse option (worse side or stimulus in mice and monkeys, respectively) and when no reward was obtained.

As expected, there were significant correlations between the proposed entropy-based metrics (see lower right of matrices in Supplementary Fig. 2). For instance, ERDS and EODS were highly correlated (Pearson: mice: $r = 0.90, p < 10^{-300}$; monkeys: $r = 0.94, p < 10^{-300}$). EODS and ERODS were also highly correlated as expected (Pearson: mice: $r = 0.95, p < 10^{-300}$; monkeys: $r = 0.97, p < 10^{-300}$). Additionally, many entropy-based metric decompositions had similarly large correlations with other entropy-based metric decompositions. Finally, we found consistent results for the relationships between undermatching and our metrics for both reward schedules, even though undermatching and our metrics were sensitive to reward probabilities on the two stimuli or actions (see Supplementary Figs. 1, 3, 4, 5).

**Entropy-based metrics can accurately predict deviation from matching**. To verify the utility of entropy-based metrics in predicting deviation from matching, we performed additional stepwise regressions to predict deviation from matching using our entropy-based metrics. In these models, we included $ERDS_+$, $ERDS_-$,

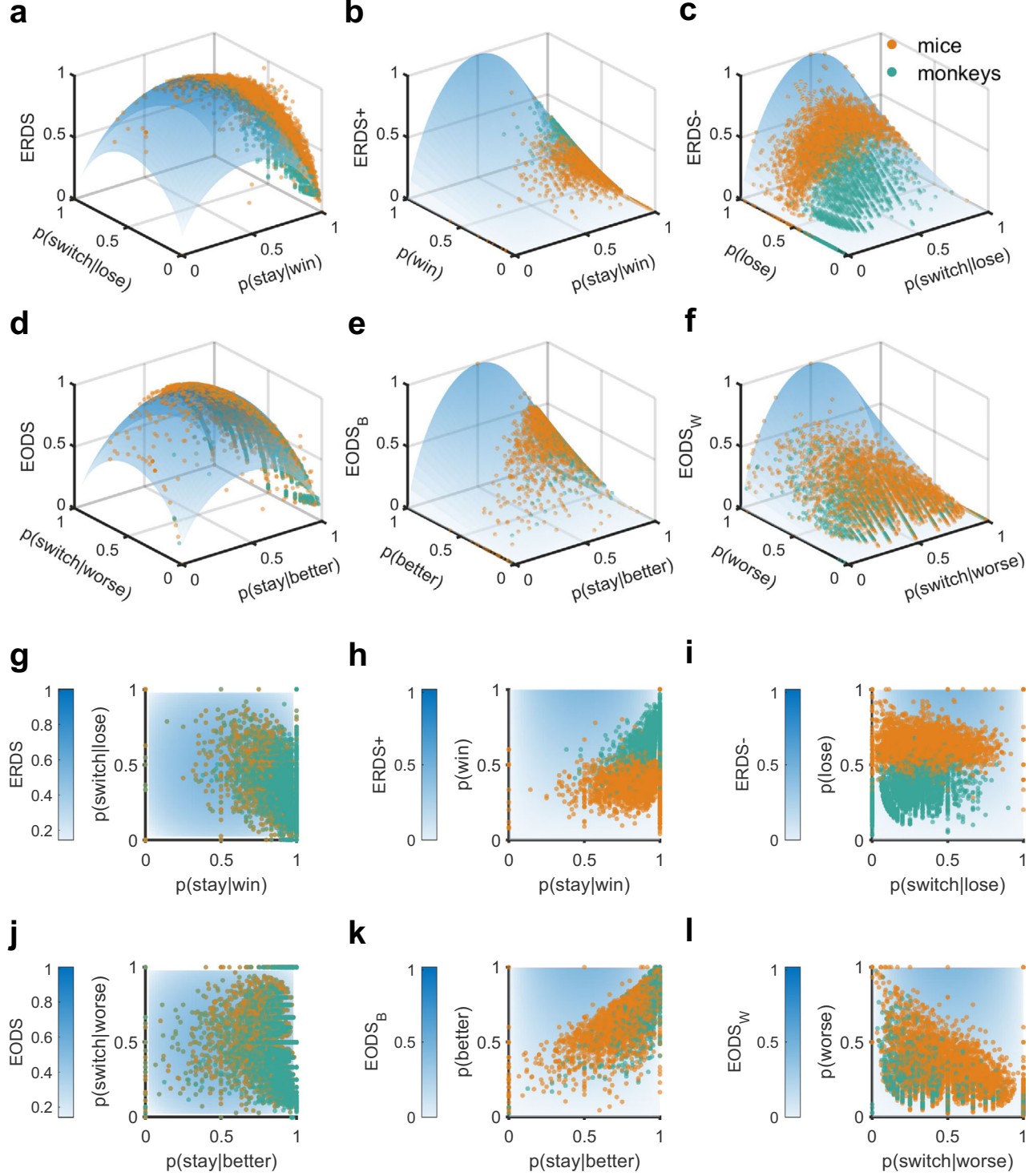

**Fig. 3 Relationship between entropy-based metrics and win-stay, lose-switch strategies. a–c** Plotted are ERDS and ERDS decompositions for rewarded and unrewarded trials (ERDS+ and ERDS−) as a function of p(win), p(lose), win-stay, and lose-switch. Darker colors correspond to larger values of metrics. For the plot in (**a**), p(win) is set to 0.5. Observed entropy-based metrics and constituent probabilities for each block for mice (orange dots) and monkeys (green dots) are superimposed on surfaces. **d–f** EODS and EODS decompositions for the better and worse options (EODS$_B$ and EODS$_W$) as a function of the probabilities of choosing the better and worse options, p(better) and p(worse), conditional probability of stay on the better option, and conditional probability of switch from the worse option. For the plot in (**d**), p(better) is set to 0.5. For all plots, the units of entropy-based metrics are bits. **g–i** Same as in (**a–c**) but using heatmap. **j–l** Same as in (**d–f**) but using heatmap.

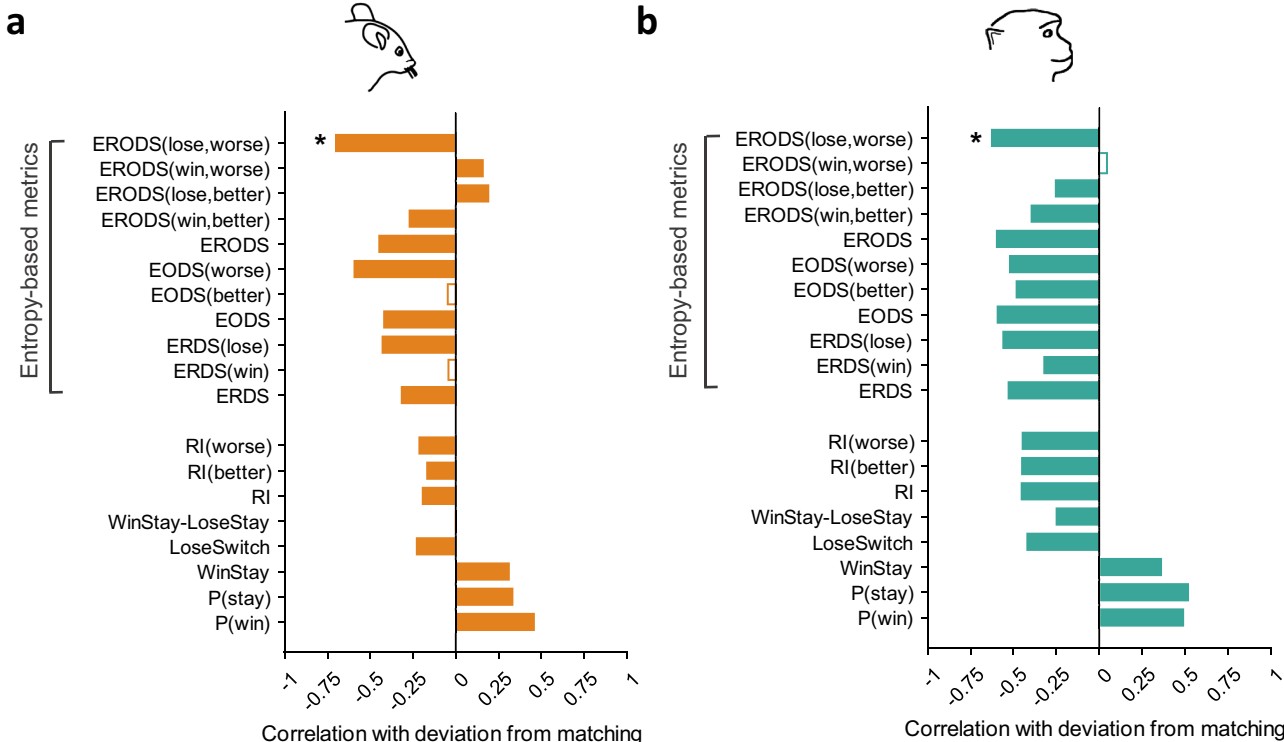

**Fig. 4 Correlation between undermatching and proposed entropy-based metrics and underlying probabilities. a** Pearson correlation between proposed entropy-based metrics and existing behavioral metrics and deviation from matching in mice. Correlation coefficients are computed across all blocks, and metrics with nonsignificant correlations (two-sided, $p > 0.0001$ to account for multiple comparisons) are indicated with a hollow bar. The metric with the highest correlation with deviation from matching is indicated with a star ($\text{ERODS}_{W-}$; $r = -0.71, p < 10^{-300}$). **b** Similar to (**a**) but for monkeys ($\text{ERODS}_{W-}$; $r = -0.64, p = 10^{-231}$). Overall, entropy-based metrics show stronger correlation with deviation from matching than existing metrics.

$EODS_B$, $EODS_W$, $ERODS_{B+}$, $ERODS_{B-}$, $ERODS_{W+}$, $ERODS_{W-}$, $RI_B$, $RI_W$, $p(win)$, $p(stay)$, $win\text{-}stay$, and $lose\text{-}switch$ as predictors (see Supplementary Note 1 for regression equations).

These models explained 74% and 57% of total variance in deviation from matching for mice and monkeys, respectively (mice: *Adjusted* $R^2 = 0.74$; monkeys : *Adjusted* $R^2 = 0.57$). For mice, the regression model explained 26% more variance than the model with repetition indices and other existing behavioral metrics and 43% more variance than the model with existing behavioral metrics but without repetition indices. For monkeys, the regression model explained 8% more variance than the model with repetition indices and existing behavioral metrics and 23% more variance than the model with existing behavioral metrics without repetition indices. These are significant improvements over previous models, suggesting that most variance in undermatching behavior can be explained by trial-by-trial response to reward feedback.

In terms of the predictive power of different metrics, we found that for mice, the first three predictors added to the regression models were $\text{ERODS}_{W-}$ ($\Delta R^2 = 0.59$), $\text{ERODS}_{W+}$ ($\Delta R^2 = 0.04$), and $\text{ERODS}_{B+}$ ($\Delta R^2 = 0.02$). For monkeys, the first three predictors added were $\text{ERODS}_{W-}$ ($\Delta R^2 = 0.31$), $\text{EODS}_W$ ($\Delta R^2 = 0.09$), and $\text{ERODS}_{B+}$ ($\Delta R^2 = 0.06$). These results indicate that entropy-based metrics were the best predictors of deviation from matching when considering all metrics together. In addition, most entropy-based metrics included as predictors were added to the final regression equations for both mice and monkeys. This suggests that despite their overlap, each entropy-based metric captures a unique aspect of the variance in deviation from matching behavior.

**Entropy-based metrics capture the relationship between undermatching and reward environment better than existing**

**metrics**. Our previous observation that entropy-based metrics can explain most variance in undermatching behavior suggests that entropy-based metrics may also capture average differences in undermatching between reward schedules in mice and the lack of differences in undermatching between reward schedules in monkeys.

To test whether this was the case, we used tenfold cross-validated linear regression to predict deviation from matching using a model without entropy-based metrics or repetition indices, a model without entropy-based metrics, and a model with all metrics (full model). Predictors chosen for inclusion in these models were the predictors that remained in final stepwise regression equations described above.

For mice, in all three models, predicted deviation from matching was significantly lower in the 40/5 than the 40/10 reward environment (Supplementary Fig. 6; two-sided $t$ test; model without entropy-based metrics or repetition indices: $p = 6.30 \times 10^{-3}$; model without entropy-based metrics: $p = 3.22 \times 10^{-3}$; full model: $p = 4.76 \times 10^{-25}$). Importantly, the difference between deviation from matching in the two reward schedules was greatest for the full model (Supplementary Fig. 6; Cohen's $d$; model without entropy-based metrics or repetition indices: $d = -0.10$; model without entropy-based metrics: $d = -0.10$; full model: $d = -0.39$). The full model with entropy-based metrics was the only model that came close to replicating the magnitude of differences in deviation from matching between the 40/5 and 40/10 schedules in behavioral data (Supplementary Fig. 6c, d).

For monkeys, predicted deviation from matching from both regression models without entropy-based metrics was significantly lower in the 70/30 than the 80/20 reward environment (Supplementary Fig. 6e, f; two-sided $t$ test; model without entropy-based metrics or repetition indices: $p = 1.19 \times 10^{-49}$;

model without entropy-based metrics: $p = 1.32 \times 10^{-29}$). Only the regression model with entropy-based metrics replicated the observed lack of difference in undermatching between reward schedules (Supplementary Fig. 6g, h; full model: $p = 0.36$; observed difference between reward schedules: $p = 0.19$). Therefore, entropy-based metrics are necessary and sufficient to capture the influence of reward schedule on deviation from matching.

**Purely RL models do not capture the pattern of entropy-based metrics.** To capture the observed variability in entropy-based metrics and underlying learning and choice mechanisms, we next fit choice behavior using three purely RL models. These models assumed different updating of reward values (RL1 and RL2; see Methods for more details) or learning multiple reward values across different timescales (multiple timescales model). We tested these models because previous research has suggested they can replicate matching or undermatching phenomena[14,26,31]. Out of these three models, we found that RL2, in which the estimated reward value for the unchosen option (side or stimulus) decays to zero over time, provided the best fit of choice behavior for both mice and monkeys as reflected in the lowest Akaike Information Criterion (AIC) (Fig. 5a, b, Supplementary Table 2).

We next tested whether RL2 could replicate observed distributions of entropy-based metrics and undermatching by simulating the model during our experiment using parameters obtained from model fitting. Due to the large number of simulations performed (100 simulations per session, mice: $n = 331,900$ blocks; monkeys: $n = 221,200$ blocks), we were able to estimate population distributions of metrics for each model. We found that the median predicted $ERODS_{W-}$ was significantly higher than the median observed $ERODS_{W-}$, suggesting the RL2 model under-utilizes loss-dependent and option-dependent strategies when compared to mice and monkeys in our experiments (Fig. 5c, d). To evaluate the similarity of observed and predicted distributions of entropy-based metrics and matching, we computed Kolmogorov's $D$ statistic that measures the maximum difference (or distance) between two empirical cumulative distribution functions. Using this method, we ound that the distribution of predicted $ERODS_{W-}$ was very different than the observed distributions for both mice and monkeys (Fig. 5c, d; two-sided Kolmogorov–Smirnov test; mice: $D = 0.121, p = 1.44 \times 10^{-41}$; monkeys: $D = 0.072, p = 1.42 \times 10^{-9}$). Moreover, the predicted distribution of deviation from matching was very different from the observed distribution for both mice and monkeys (Fig. 5e, f; Two-sided Kolmogorov–Smirnov test; mice: $D = 0.091$, $p = 3.24 \times 10^{-24}$; monkeys: $D = 0.101, p = 6.38 \times 10^{-20}$).

Finally, we also computed undermatching and all behavioral metrics in simulated data using RL2 with random parameter values. We found that our entropy-based metrics were better predictors for deviation from matching than the parameters of the RL2 model (see Supplementary Fig. 7). Together, our results illustrate that purely RL models fail to replicate observed distribution of $ERODS_{W-}$ and variability in matching behavior, pointing to additional mechanisms that contribute to behavior. Moreover, $ERODS_{W-}$ was highly correlated with undermatching in observed behavior (Fig. 4) and RL simulations (Supplementary Fig. 7), suggesting that a model that better captures $ERODS_{W-}$ may also better capture variability in matching behavior.

**Model with additional choice memory captures entropy-based metrics in monkeys more accurately.** The deviations of predicted $ERODS_{W-}$ from observed $ERODS_{W-}$ suggest RL models underutilize loss-dependent and option-dependent strategies; that is, they fail to capture the influence of option (stimulus or action) and loss in the current trial on choice in the subsequent trial. To improve capture of option-dependent strategy, we added a common choice-memory component to estimate the effects of previous choices on subsequent decisions[8,15,38]. The choice-memory (CM) component encourages either staying on or switching from options that have been chosen recently. Because standard RL models typically choose the option with a higher value, the CM component can capture strategy in response to selection of the better or worse option reflected in the option-dependent entropy-based metrics. The influence of the CM component on choice is determined by fitting a weight parameter that can take either positive or negative values which correspond to better-stay/worse-switch or better-switch/worse-stay strategies, respectively.

In monkeys, we found that the RL2 model augmented with a CM component, which we refer to as the RL2+CM model, fit choice behavior better than RL1, RL2, and RL1 + CM as indicated by lower AIC (Fig. 5b; Supplementary Table 2). Although the improvement in fit of choice behavior for RL2 + CM over RL2 was statistically significant (paired samples $t$ test of AICs: $p = 1.04 \times 10^{-23}$; Supplementary Table 2), the RL2 + CM model was only twice as likely as RL2 to be the best model based on a comparison of Akaike weights.

Importantly, the RL2 + CM model improved capture of the observed distribution of $ERODS_{W-}$ in monkeys (Fig. 5d; two-sided Kolmogorov–Smirnov test; $D = 0.037, p = 8.91 \times 10^{-3}$). This improvement in capturing $ERODS_{W-}$ corresponded with similar improvements in capturing deviation from matching. The predicted distribution of deviation from matching from the RL2 + CM model better replicated the observed distribution of deviation from matching than the predicted distribution from RL2 (Fig. 5f; two-sided Kolmogorov–Smirnov test; $D = 0.065, p = 2.07 \times 10^{-8}$). This improvement was significant; there was an over 30% reduction in the maximum difference between CDFs in the RL2 + CM model from the RL2 model.

To determine whether these improvements were attributable to modulations in better-switch/worse-stay or better-stay/worse-switch strategies, we examined the distribution of the estimated CM weights and fit a model with a CM component with weights restricted to positive values only (RL2 + CM + model). The median fitted CM weights in the RL2 + CM model was negative (Supplementary Fig. 8k), and the fit of choice behavior was worse for the RL2 + CM + model than the RL2 + CM model (Supplementary Table 2), indicating that the CM component enhanced better-switch/worse-stay strategies in monkeys.

In mice however, the RL2 + CM model had positive weights and had fairly weak effects on fit of local choice behavior and capture of metrics and undermatching (Supplementary Figs. 8i, 5a; Supplementary Table 2). These results in conjunction with our entropy-based findings suggest that additional mechanisms that modulate response to loss are necessary to improve capture of variability in $ERODS_{W-}$ and matching behavior in mice.

**New model with additional choice and loss memories captures choice behavior and entropy-based metrics in mice more accurately.** To better capture loss-dependent strategy in mice, we augmented the RL2 + CM model with a new outcome-dependent loss-memory component (see Methods). The loss-memory (LM) component encourages either staying or switching in response to loss (increases lose-switch or lose-stay) and is modulated by expected uncertainty. Here, expected uncertainty is defined as the expected unsigned reward prediction error in the RL2 component. As uncertainty increases, the weight of the LM component increases proportionately, making choice more dependent on feedback in the previous trial. In contrast to RL models with adaptive learning rates (e.g., as in Pearce-Hall model), changes in

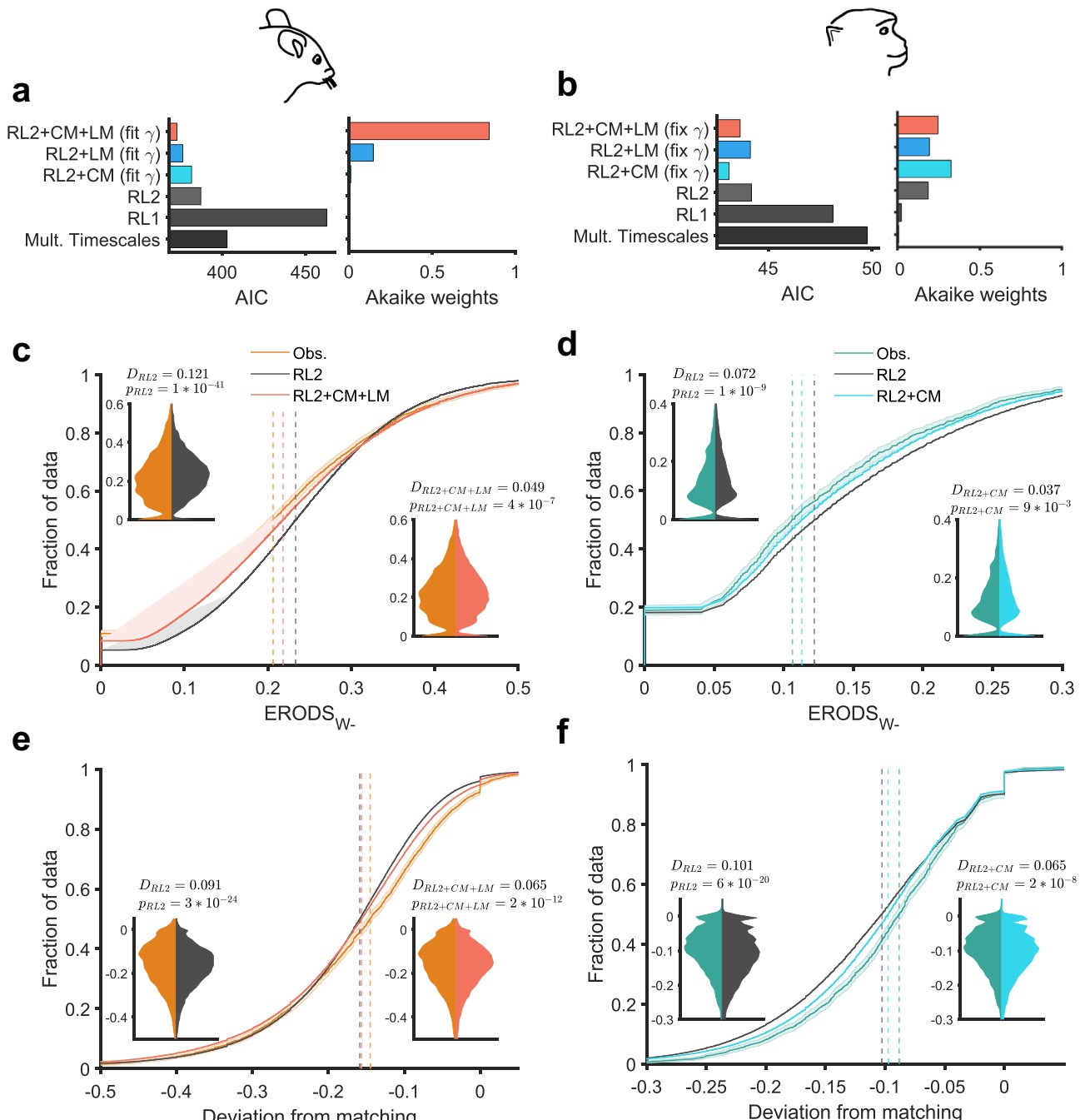

**Fig. 5 RL2 + CM + LM and RL2 + CM models better account for choice behavior, undermatching, and entropy-based metrics in mice and monkeys, respectively. a, b** Comparison of goodness-of-fit of a return-based (RL1) model, income-based (RL2) model, income-based models augmented with choice memory (CM) and/or loss-memory (LM) components, and a model based on learning on multiple timescales. Plotted is the Akaike Information Criterion (AIC) averaged over all sessions and Akaike weights computed with the average AIC for mice (**a**) and monkeys (**b**). **c–f** Empirical cumulative distribution functions of $ERODS_{W-}$ (**c, d**) and deviation from matching (**e, f**) observed in animals and predicted from simulations of the RL2 and RL2 + CM + LM models. Shaded bars around CDFs indicate 95% confidence interval. Dashed vertical lines indicate the median of each distribution. Insets display the distribution of observed metrics versus metrics predicted using the RL2 model (left inset) and RL2 + CM + LM model and RL2 + CM model (right inset) for mice and monkeys, respectively. Displayed $D$-values and $p$ values are the test-statistic and $p$ value from a two-sided Kolmogorov–Smirnov test comparing the distributions. The RL2 + CM + LM model and RL2 + CM better captured deviation from matching by over 20% and over 30% in mice and monkeys, respectively.

the LM component only influence subsequent choice, not the aggregation of values in the RL component of the model. The model is also agnostic about the strategy employed by the LM component such that it can encourage lose-stay or lose-switch behavior. The addition of the LM component to the RL + CM model creates a model that can flexibly modulate loss-dependent

strategies, which may increase the consistency of strategy in response to loss on the worse option (reduce $ERODS_{W-}$).

In mice, the RL2 + CM + LM model (RL2 augmented with a choice-memory and a loss-memory component) fit choice behavior better than all existing RL models as indicated by a lower AIC (Fig. 5a). The Akaike weight for the RL2 + CM + LM

model was 0.84, which suggests there is a high probability that the RL2 + CM + LM model is the best model out of all models examined. The RL2 + CM + LM model also captured the observed distribution of $ERODS_{W-}$ for mice better than RL2 (Fig. 5c, d; two-sided Kolmogorov–Smirnov test; $D = 0.049, p = 3.77 \times 10^{-7}$). Moreover, the predicted distribution of deviation from matching from the RL2 + CM + LM model better replicated the observed distribution of deviation from matching than the predicted distribution from RL2 (Fig. 5e; two-sided Kolmogorov–Smirnov test; mice: $D = 0.065, p = 2.19 \times 10^{-12}$). This improvement corresponds to an over 20% reduction in the maximum difference between cumulative distribution functions (CDFs) for deviation from matching computed from observed and simulated data.

Finally, to understand how the RL2 + CM + LM model modulates specific loss- and option-dependent strategies in mice, we compared the distributions of model parameters between all models with RL2. The LM and CM components both had positive average weights in mice (Supplementary Fig. 8i–l), such that the LM component encouraged lose-switch strategies and the choice-memory component encouraged worse-switch strategies. Two additional models, RL2 + CM+ and RL2 + LM+, in which the weights of the CM and LM components were restricted to positive values, fit comparably to the RL2 + CM and RL2 + LM models, providing further evidence for the previous conclusion. Interestingly, the average weights of the CM and LM components were higher in the RL2 + CM + LM model than in the RL2 + CM and RL2 + LM models, indicating that the two components may interact to modulate behavior (Supplementary Fig. 8i, j). This dovetails with our entropy-based results that show response to the loss after selection of the worse option is uniquely important for global choice behaviors.

In monkeys, however, the LM component was not necessary to better explain choice behavior. The RL2 + CM model fit local choice behavior better than the RL2 + CM + LM model based on AIC and better captured $ERODS_{w-}$ and deviation from matching (Fig. 5a; Supplementary Table 2). Despite this, in monkeys, the RL2 + LM model still improved upon RL2 in capture of undermatching and $ERODS_{W-}$ (Supplementary Table 2), indicating that there may be overlap between the effects of the CM and LM components that renders the LM component useless in the full model.

To summarize, the models with additional components improved fit of choice behavior and captured our metrics and undermatching more accurately in both species, whereby revealing that undermatching behavior arises from competition among multiple components incorporating choice memory and/or loss memory on a trial-by-trial basis. Importantly, we used deviations in predicted entropy-based metrics from their observed values to identify shortcomings in purely RL models and to incorporate previous mechanisms or propose new mechanisms to mitigate them.

## Discussion

Undermatching is a universal behavioral phenomenon that has been observed across many species. Here, we show that proposed entropy-based metrics based on response to reward feedback can accurately predict undermatching in mice and monkeys, suggesting that inconsistencies in the use of local reward-dependent and option-dependent strategies can account for a large proportion of variance in global undermatching. Moreover, we demonstrate that these entropy-based metrics can be utilized to construct more complex RL models that are able to capture choice behavior, undermatching, and utilization of reward-dependent strategies. Together, our entropy-based metrics

provide a model-free tool to develop and refine computational models of choice behavior and reveal neural mechanisms underlying adaptive behavior.

Similar to many previous studies of matching behavior[4,11,14,17,39], we observed significant, but highly variable undermatching in both mice and monkeys. By focusing on the variability in undermatching, here, we were able to show that global undermatching can be largely explained by the degree of inconsistency in response to no reward on the worse option ($ERODS_{w-}$) across species. Specifically, $ERODS_{w-}$ could explain about 50% and 41% of the variance in undermatching in mice and monkeys, respectively. The proposed entropy-based metrics were able to predict undermatching across two very different species despite differences in the tasks including utilized reward probabilities and schedules (40/5 and 40/10 probabilities with baiting vs. complementary 80/20 and 70/30 with no baiting), learning modality (action-based vs. stimulus-based), choice readout (licks vs. saccades), and predictability of block switches (unpredictable vs. semipredictable), suggesting the proposed metrics are generalizable.

The proposed entropy-based metrics complement and improve upon commonly used behavioral metrics such as win-stay, lose-switch, and the U-measure[40]. Although win-stay and lose-switch provide valuable information[28,41–45], these probabilities do not solely reflect the effects of reward feedback on staying (or similarly switching) as they both depend on the probability of stay. For example, if staying behavior is independent of reward, win-stay, and lose-switch values simply reflect the overall stay and switch probabilities, respectively. Consistently, we found that win-stay and lose-switch are not strong predictors of undermatching because their relative importance depends on the overall probability of winning. For example, if the overall probability of reward is high, lose-switch is less useful for predicting behavior because response to loss represents strategy in a small subset of trials. Therefore, win-stay and lose-switch cannot capture the degree to which staying and switching strategies depend on reward outcome only. The entropy-based metrics such as ERDS overcome these issues by combing win-stay and lose-switch with p(win) and p(stay). Similarly, although the U-value has been used to measure consistency or variability in choice behavior[46,47], this metric is difficult to interpret and fails to capture sequential dependencies in choice[48]. Our proposed entropy-based metrics avoid these issues because they have both clear interpretations and can capture the sequential dependence of choice on previous reward and/or selected action or option.

As shown by multiple studies, models that fit choice data best may still fail to replicate important aspects of behavior[33,34]. Therefore, model validation must involve analyzing both a model's predictive potential (fitting) and its generative power (replication of behavior in simulations). We used shortcomings of purely RL models in capturing the most predictive entropy-based metrics to detect additional mechanisms underlying adaptive behavior. This approach can be applied to other tasks in which similar or different entropy-based metrics are most predictive of global choice behavior (matching or other metrics). Our aim here was not to find the best model for capturing all aspects of behavior but instead, to provide a framework for how local response to reinforcement can be used to guide model development and explore interesting properties of local and global choice behavior.

Using this method, we constructed a model (RL2 + CM + LM model) that augments a reinforcement-learning model with a choice-memory component that captures option-dependent strategies and a loss-memory component that captures loss-dependent strategies. Previous studies have also shown that a combination of WSLS strategies with RL models could improve

fit of choice behavior and capture of the average matching behavior[27,28,49]. The choice-memory component used here is similar to other choice-memory components that have been shown to improve fit of choice behavior[15,38]. Nonetheless, the proposed RL2 + CM + LM model is a novel combination of these components. Critically, the weights of the loss and choice components could be either positive or negative. This parallels how entropy-based metrics capture response to reward feedback considering that low entropy can result from strong positive or negative influences of recent rewards or choices (e.g., high win-stay and high win-switch both correspond to low entropy). Neural correlates of a similar loss-memory component weighted by recent reward prediction errors have been identified in the dorsal anterior cingulate cortex of humans[50]. Moreover, neural correlates of such choice memories have been identified in various cortical areas of monkeys including the dorsolateral prefrontal cortex, dorsal medial prefrontal cortex, lateral intraparietal area, and the anterior cingulate cortex[51–53].

Despite the significant correlation between ERODSW_ and deviation from matching in both species, the loss-memory component introduced here only improved fit of choice behavior and capture of metrics in the full model in mice. This finding may be related to the close correspondence between reward- and option-dependent strategies in the monkey task because winning (respectively, losing) almost always corresponds with choosing the better (respectively, worse) side. Due to this significant overlap, one component may be sufficient to capture both strategies. In the mouse task, however, these strategies were dissociated because losing was likely when choosing either the better or worse option (but more for the worse option). This could explain why for monkeys, the LM component improved capture of entropy-based metrics and deviation from matching in the RL2 + LM model relative to the RL2 model but was not useful in conjunction with the choice-memory component. Moreover, we observed a higher overall probability of switching in mice than in monkeys, indicating that mice occasionally switch from the more-rewarding side to harvest baited rewards on the less-rewarding side, whereas monkeys typically exploit the more-rewarding stimulus. Because of this, a loss-memory component that encourages switching in response to loss would be more helpful in capturing that behavior in mice than in monkeys. Although aforementioned differences in results for these two datasets may be partially explained by differences in task structure and species, they also highlight the limitations of using entropy-based metrics to guide model development. Entropy-based metrics describe properties of choice behavior that are helpful for making educated guesses about model structure, but alone, cannot provide a generative account of behavior.

The model fits were also worse for mouse data than the monkey data in terms of explained variance in choice behavior, likely due to differences in the overall entropy in choice behavior and task structure. More specifically, mice showed higher average entropy in their choice behavior than monkeys across different measures, suggesting that the observed difference in the quality of fit occurred because mice choice behavior was more random and thus harder to predict. In addition, sessions in the mouse task were longer than superblocks in the monkey task, so the same number of parameters were used to account for more choices in mice than in monkeys, resulting in an overall poorer fitting quality.

The goal of our approach, to predict and develop generative models to explain undermatching, was similar to a recent study that suggested limited undermatching results in optimal performance in stochastic environments and proposed learning on multiple timescales to account for such undermatching[31]. In contrast, we identified a positive correlation between reward harvesting and deviation from matching which suggests that the degree of undermatching observed here corresponded with sub-optimal choice. This difference between Iigaya et al.[31] and our study could be due to differences in how performance and undermatching are defined. More specifically, here we measure performance as the total number of harvested rewards in each block of trials and undermatching as the difference between choice and reward fractions in each block. In contrast, Iigaya et al.[31] use harvesting efficiency, equal to the number of rewards harvested divided by the maximum number of rewards that could have been collected, in each session of experiment (consisting of multiple blocks) as a measure of performance and quantify undermatching as the difference between the slope of choice vs. reward fractions and one in each session. Moreover, we found that nearly all other models described here better accounted for local and global choice behavior than the multiple timescales model proposed in Iigaya et al.[31]. Nonetheless, it is possible that more complex models based on learning on multiple timescales may fit choice behavior better.

We also observed weak, positive choice-memory effects in mice such that mice tended to choose options that they had recently chosen. A previous study using a nearly identical task (reversal learning with same reward schedules (40/10) and baited rewards, but longer blocks) observed a much stronger, negative choice memory effect in mice[8]. The reason for this difference is unclear given the similarity of the two tasks. Consistent with prior studies of choice-history effects in monkeys[15], we identified strong, negative choice-memory effects in monkeys such that the choice-memory component encouraged switching from recently chosen options. Thus, the incorporation of the negative weights was only important for capturing behavior in the monkey task and thus, could be task dependent. This negative weighting mechanism may be able to facilitate quick adaptation to reversals in monkeys, a behavior that has previously been described using a Bayesian approach[54], because negative weights in either the choice-memory or the loss-memory component encourage faster response to reversals. Future studies are needed to test whether this is the case.

In summary, we show that entropy-based metrics are good predictors of global choice behavior across species and can be used to refine RL models. Results from fitting and simulating RL models augmented with additional components suggest that recent choices and rewards affect decisions in ways beyond their influence on the update of subjective values in standard RL models. Thus, entropy-based metrics have the potential to open a realm of possibilities for understanding computational and neural mechanisms underlying adaptive behavior.

## Methods

**Experimental paradigm in mice**. Mice performed a dynamic foraging task in which after receiving a go-cue signaled by an odor, they licked one of the two water tubes (on left and right) to harvest possible reward. In 5% of trials, a no-go cue was presented by another odor signaling that a lick would not be rewarded or punished. If a mouse licked one of the tubes after a go-cue odor, reward was delivered probabilistically. Each trial was followed by an inter-trial interval drawn from an exponential distribution with a rate parameter of 0.3. If a mouse licked a tube in the 1 s no-lick prior to odor delivery, an additional inter-trial interval and an additional 2.5 s no-lick period were added.

The reward probabilities assigned to the left and right tubes were constant for a fixed number of trials (blocks) and changed throughout the session (block switches). Block lengths were drawn from a uniform distribution that spanned a range of 40–100 trials, however, the exact block lengths spanned smaller ranges for individual sessions, resulting in variable block lengths with most block lengths ranging between 40 and 80 trials. If mice exhibited strong side-specific biases, block lengths were occasionally shortened or lengthened. Miss trials, in which the mouse did not make a choice, and no-go trials were excluded for all analyses described here. In total, 1706 miss trials (average of 3.64 per session) and 7893 no-go trials (average of 16.83 trials per session) were excluded from our analyses.

Mice performed two versions of the task, one with 16 different sets of reward schedules and another with two sets of reward schedules. The vast majority (469 out of 528) of sessions used two sets of reward probabilities equal to 0.4 and 0.1, and 0.4 and 0.05, which we refer to as 40/10 and 40/5 reward schedules. Here, we focus on the most frequent blocks (40/5 and 40/10 reward schedules). Rewards were baited such that if reward was assigned on a given side and that side was not selected, reward would remain on that side until the next time that side was selected. Due to this baiting mechanism, the probability of obtaining reward on the unchosen side increased over time as during foraging in a natural environment. In total, 16 mice performed 469 sessions of the two-probability version of the task for a total of 3319 blocks (1786 and 1533 blocks with 40/5 and 40/10 reward schedules, respectively) and 189,199 trials. Male C57BL/6J (The Jackson Laboratory, 000664) mice were used in the experiment, and mice were housed on a 12 h dark/12 h light cycle. All surgical and experimental procedures were in accordance with the National Institutes of Health Guide for the Care and Use of Laboratory Animals and approved by the Johns Hopkins University Animal Care and Use Committee. This experimental setup and some analyses of the data have also been described in Bari et al.[9].

**Experimental paradigm in monkeys.** In the reversal learning task in monkeys, Costa et al.[35] trained monkeys to fixate on a central point on a screen to initiate each trial (Fig. 1). After fixation, two stimuli, a square and circle, were presented on the screen to the left and right of the fixation point (6° visual angle). The side that the stimuli were presented on was assigned randomly and was not related to reward. Monkeys made saccades to a stimulus and fixated on the stimulus to indicate their choice in each trial. A 0.085 mL juice reward was delivered probabilistically via a pressurized tube based on the chosen stimulus. Each trial was followed by a fixed 1.5 s inter-trial interval. Trials in which the monkey did not make a choice or failed to fixate were immediately repeated.

Monkeys completed sessions that contained around 1300 trials on average divided into superblocks of 80 trials. Within each superblock the reward probabilities assigned to each cue were reversed randomly between trials 30 and 50, such that the stimuli that was less rewarding at the beginning of the superblock became more rewarding and vice versa. Every 80 trials, monkeys were presented with new stimuli that varied in color but not shape. Six images of a red, green, and blue circle or square were used as stimuli, and the two choice options in a given block always differed in both color and shape, e.g., a red square could be presented with a blue circle. Superblock presentation was fully randomized without replacement such that a monkey viewed all stimuli pair/reward schedule combinations (e.g., red square, blue circle, 70/30) before any repeated.

Monkeys performed two variants of the task, a stochastic variant with three reward schedules (80/20, 70/30, and 60/40) and a deterministic variant with one reward schedule (100/0). Here, we focus our analyses on the 80/20 and 70/30 reward schedules (2212 blocks of the task performed by 4 monkeys) as they provide two levels of uncertainty similar to the experiment in mice. Male rhesus macaques were used in the experiment. Monkeys were water restricted throughout the experiment and during test days earned fluid only through the task. Stimulus presentation and behavioral monitoring was controlled by the MonkeyLogic (version 1.1) toolbox[55]. Eye movements were sampled at 1 kHz using an Arrington eye-tracking system (Arrington Research). All experimental procedures were performed in accordance with the Guide for the Care and Use of Laboratory Animals and were approved by the National Institute of Mental Health Animal Care and Use Committee. This experimental setup and some analyses of the its data have also been described in Costa et al.[35,54].

**Behavioral metrics**

*Matching performance.* To measure the overall response to reinforcement on the two choice options (e.g., left and right actions when reward is based on the location) in each block of the experiment, we defined undermatching (UM) as:

$$UM = (Choice_F - Reward_F) \times sign(Reward_F - 0.5) \quad (1)$$

where $sign(0) = 1$ and choice and reward fractions ($Choice_F$, $Reward_F$) are defined as follows:

$$Choice_F = \frac{P(choosing\ left)}{P(choosing\ left) + P(choosing\ right)}$$

$$Reward_F = \frac{P(reward\ on\ left)}{P(reward\ on\ left) + P(reward\ on\ right)} \quad (2)$$

Therefore, UM measures the difference between choice and reward fractions toward the more-rewarding side. Similarly, UM can be computed based on the color of stimuli when color is informative about reward outcome. Based on our definition, negative and positive values for UM correspond to undermatching and overmatching, respectively.

*Win-stay and lose-switch.* Win-stay (WS) and lose-switch (LS) measure the tendency to repeat a rewarded choice (in terms of action or stimulus) and switch away from an unrewarded choice, respectively. These quantities are based on the conditional probabilities of stay and switch after reward and no reward, respectively,

and can be calculated in a block of trials as follows:

$$WS = P(stay|win) = \frac{P(stay, win)}{P(win)}$$

$$LS = P(switch|lose) = \frac{P(switch, lose)}{P(lose)} \quad (3)$$

where $P(win)$ and $P(stay)$ are the probabilities of harvesting reward and choosing the same option (side or stimulus) in successive trials, $P(lose) = 1 - P(win)$, and $P(switch) = 1 - P(stay)$.

When computing metrics based on action or reward in the previous trial for mice, we treated each miss trial as though the trial did not exist. For example, if a mouse chose left and was rewarded on trial $t$, did not respond on trial $t + 1$ (miss trial), then chose left on trial $t + 2$, trial $t + 2$ would be labeled as win-stay.

*Repetition index.* Repetition index (RI) measures the tendency to repeat a choice beyond what is expected by chance and can be computed by subtracting the probability of stay by chance from the original probability of stay[36]. RI can be computed based on the repetition of left or right choices ($RI_{LR}$) as follows:

$$RI_{LR} = P(stay) - (P(left) \times P(left) + P(right) \times P(right)) \quad (4)$$

In general, RI reflects a combination of reward-dependent and reward-independent strategies as well as the sensitivity of choice to value differences (equal to the inverse temperature in the logit function translating value differences to choice probability; see Eq. (28)).

Repetition index can also be measured based on other option or choice attributes that predict reward such as the color of the chosen option. For example, RI can be defined based on selection of the better or worse option ($RI_{BW}$) when such options exist in a task:

$$RI_{BW} = P(stay) - (P(better) \times P(better) + P(worse) \times P(worse))$$
$$= (P(better(t), better(t - 1)) - P(better) \times P(better)) \quad (5)$$
$$+ (P(worse(t), worse(t - 1)) - P(worse) \times P(worse))$$

where $t$ is the trial number. Using Eq. (5), $RI_{BW}$ can be decomposed into two pieces, $RI_B$ and $RI_W$, that measure the tendency to repeat the better and worse options, respectively:

$$RI_B = P(better(t), better(t - 1)) - P(better) \times P(better)$$

$$RI_W = P(worse(t), worse(t - 1)) - P(worse) \times P(worse) \quad (6)$$

*Entropy-based metrics.* In order to quantify the influence of previous reward outcome on choice behavior in terms of stay or switch, we defined the conditional entropy of reward-dependent strategies (ERDS) that combines tendencies of win-stay and lose-switch into a single metric. More specifically, ERDS is defined as the conditional entropy of using stay or switch strategy depending on win or lose in the preceding trial:

$$ERDS = H(str|rew) = -\left( P(stay, win) \times \log_2\left(\frac{P(stay, win)}{P(win)}\right) \right.$$
$$+ P(switch, win) \times \log_2\left(\frac{P(switch, win)}{P(win)}\right) + P(stay, lose) \quad (7)$$
$$\times \log_2\left(\frac{P(stay, lose)}{P(lose)}\right) + P(switch, lose) \times \log_2\left(\frac{P(switch, lose)}{P(lose)}\right) \right)$$

To better show the link between ERDS and win-stay, lose-switch, and p(win), Eq. (7) can be rewritten as follows:

$$ERDS = -(P(win) \times WS \times \log_2(WS) + P(win) \times (1 - WS)$$
$$\times \log_2(1 - WS) + (1 - P(win)) \times (1 - LS) \times \log_2(1 - LS) \quad (8)$$
$$+ (1 - P(win)) \times LS \times \log_2(LS))$$

ERDS can be decomposed into two components, $ERDS_+$ and $ERDS_-$ ($ERDS = ERDS_+ + ERDS_-$), to allow separation of animals' response to rewarded (win) and unrewarded (loss) outcomes:

$$ERDS_+ = H(str|win) = -\left( P(stay, win) \times \log_2\left(\frac{P(stay, win)}{P(win)}\right) \right.$$
$$\left. + P(switch, win) \times \log_2\left(\frac{P(switch, win)}{P(win)}\right) \right) \quad (9)$$

$$ERDS_- = H(str|lose) = -\left( P(stay, lose) \times \log_2\left(\frac{P(stay, lose)}{P(lose)}\right) \right.$$
$$\left. + P(switch, lose) \times \log_2\left(\frac{P(switch, lose)}{P(lose)}\right) \right) \quad (10)$$

The above equations also show that $ERDS_+$ and $ERDS_-$ are linked to win-stay and lose-switch, respectively.

Considering that RI can be decomposed to repetition after the better or worse option (Eq. (5)), and following the same logic used to derive ERDS, one can define the conditional entropy of option-dependent strategy (EODS) based on staying on

or switching from the better or worse option ($EODS_{BW}$) in two consecutive trials:

$$EODS_{BW} = H(str|opt) = -\left( P(stay, better) \times \log_2\left(\frac{P(stay, better)}{P(better)}\right) + P(switch, better) \right.$$
$$\times \log_2\left(\frac{P(switch, better)}{P(better)}\right) + P(stay, worse) \times \log_2\left(\frac{P(stay, worse)}{P(worse)}\right)$$
$$\left. + P(switch, worse) \times \log_2\left(\frac{P(switch, worse)}{P(worse)}\right)\right)$$
(11)

$EODS_{BW}$ can be decomposed into two components based on the better and worse options:

$$EODS_{B} = H(str|better) = -\left( P(stay, better) \times \log_2\left(\frac{P(stay, better)}{P(better)}\right) \right.$$
$$\left. + P(switch, better) \times \log_2\left(\frac{P(switch, better)}{P(better)}\right)\right)$$

$$EODS_{W} = H(str|worse) = -\left( P(stay, worse) \times \log_2\left(\frac{P(stay, worse)}{P(worse)}\right) \right.$$
$$\left. + P(switch, worse) \times \log_2\left(\frac{P(switch, worse)}{P(worse)}\right)\right)$$
(12)

EODS can also be computed based on other choice attributes (e.g., location or color of options) to examine what information drives stay or switch behavior:

$$EODS_{LR} = H(str|opt)$$
$$= -\left( P(stay, right) \times \log_2\left(\frac{P(stay, right)}{P(right)}\right) + P(switch, right) \right.$$
$$\times \log_2\left(\frac{P(switch, right)}{P(right)}\right) + P(stay, left) \times \log_2\left(\frac{P(stay, left)}{P(left)}\right)$$
$$\left. + P(switch, left) \times \log_2\left(\frac{P(switch, left)}{P(left)}\right)\right)$$
(13)

We should note that ERDS and EODS are directly comparable and provide insight into the consistency of strategy adopted by an animal. Lower ERDS than EODS suggests that an animal's decisions are more consistently influenced by immediate reward feedback than selection of the better or worse option. A lower EODS than ERDS suggests the opposite. It is worth noting that ERDS decompositions (ERDS+ or ERDS−) cannot be directly compared to EODS decompositions (EODS$_B$ or EODS$_W$) because they encompass different sets of trials; that is, trials where the animal wins may not be trials where the animal chooses the better option and vice versa.

Because conditional entropies can be defined for any two discrete random variables, ERDS and EODS can be generalized to combinations or sequences of combinations of reward and option. Hence, we can define the entropy of reward- and option-dependent strategy (ERODS), a measure of the dependence of strategy on the selected option and reward outcome.

$$ERODS = H(str|rew, opt)$$
$$= -\left( P(stay, win, better) \times \log_2\left(\frac{P(stay, win, better)}{P(win, better)}\right) \right.$$
$$+ P(stay, win, worse) \times \log_2\left(\frac{P(stay, win, worse)}{P(win, worse)}\right)$$
$$+ P(switch, win, better) \times \log_2\left(\frac{P(switch, win, better)}{P(win, better)}\right)$$
$$+ P(switch, win, worse) \times \log_2\left(\frac{P(switch, win, worse)}{P(win, worse)}\right)$$
$$+ P(stay, lose, better) \times \log_2\left(\frac{P(stay, lose, better)}{P(lose, better)}\right)$$
$$+ P(stay, lose, worse) \times \log_2\left(\frac{P(stay, lose, worse)}{P(lose, worse)}\right)$$
$$+ P(switch, lose, better) \times \log_2\left(\frac{P(switch, lose, better)}{P(lose, better)}\right)$$
$$\left. + P(switch, lose, worse) \times \log_2\left(\frac{P(switch, lose, worse)}{P(lose, worse)}\right)\right)$$
(14)

ERODS can be decomposed based on choosing the better or worse option in the previous trial, winning or losing in the previous trial, or combinations of the selected option and reward outcome (e.g., choose better option and win on the previous trial).

Decomposing ERODS based on reward option combinations gives:

$$ERODS_{B+} = H(str|win, better)$$
$$= -\left( P(stay, win, better) \times \log_2\left(\frac{P(stay, win, better)}{P(win, better)}\right) \right.$$
$$\left. + P(switch, win, better) \times \log_2\left(\frac{P(switch, win, better)}{P(win, better)}\right)\right)$$

$$ERODS_{W+} = H(str|win, worse)$$
$$= -\left( P(stay, win, worse) \times \log_2\left(\frac{P(stay, win, worse)}{P(win, worse)}\right) \right.$$
$$\left. + P(switch, win, worse) \times \log_2\left(\frac{P(switch, win, worse)}{P(win, worse)}\right)\right)$$

$$ERODS_{B-} = H(str|lose, better)$$
$$= -\left( P(stay, lose, better) \times \log_2\left(\frac{P(stay, lose, better)}{P(lose, better)}\right) \right.$$
$$\left. + P(switch, lose, better) \times \log_2\left(\frac{P(switch, lose, better)}{P(lose, better)}\right)\right)$$

$$ERODS_{W-} = H(str|lose, worse)$$
$$= -\left( P(stay, lose, worse) \times \log_2\left(\frac{P(stay, lose, worse)}{P(lose, worse)}\right) \right.$$
$$\left. + P(switch, lose, worse) \times \log_2\left(\frac{P(switch, lose, worse)}{P(lose, worse)}\right)\right)$$
(15)

Finally, ERODS can also be decomposed based on selection of the better or worse option:

$$ERODS_{B} = H(str|rew, better) = ERODS_{B+} + ERODS_{B-}$$
$$ERODS_{W} = H(str|rew, worse) = ERODS_{W+} + ERODS_{W-}$$
(16)

or winning or losing in the previous trial:

$$ERODS_{+} = H(str|win, opt) = ERODS_{B+} + ERODS_{W+}$$
$$ERODS_{-} = H(str|lose, opt) = ERODS_{B-} + ERODS_{W-}.$$
(17)

**Reinforcement-learning models.** We used nine generative RL models to fit choice behavior. In all models except the multiple timescales model, reward values associated with the right and left sides ($Q_{Right}$ and $Q_{Left}$) for mice or circle and square stimuli ($Q_{Circle}$ and $Q_{Square}$) for monkeys were updated differently depending on whether a given choice was rewarded or not. Some of the models incorporated additional loss- or choice-memory components that influenced choice but did not affect the update of reward values. As such, we refer to the final reward and non-reward values used for decision making as decision values, $DV$ (e.g., $DV_{Circle}$), to distinguish them from the updated reward values. Models were defined in a nested fashion with subsequent models building on the update rules of their predecessor.

*Purely RL models.* In the first model, which we refer to as RL1, only the reward value associated with the chosen option (side or stimulus) ($Q_C^{RL1}$) was updated as follows:

$$Q_C^{RL1}(t + 1) = Q_C^{RL1}(t) + \alpha\left(R(t) - Q_C^{RL1}(t)\right)$$
(18)

where $C \in \{Left, Right\}$ for mice and $C \in \{Circle, Square\}$ for monkeys, $R(t) = 1$ or 0 indicates reward outcome on trial $t$, and $\alpha$ corresponds to the learning rate ($\alpha_{rew}$ or $\alpha_{unrew}$) depending on the whether the choice was rewarded or not rewarded. In contrast, the reward value associated with the unchosen option ($Q_U^{RL1}$) was not updated in this model:

$$Q_U^{RL1}(t + 1) = Q_U^{RL1}(t)$$
(19)

where $U \in \{Left, Right\}$ for mice and $U \in \{Circle, Square\}$ for monkeys. In RL1, $DV_i = Q_i^{RL1}$.

In the second model (RL2), the reward value associated with the chosen option ($Q_C^{RL2}$) was updated as in Eq. (18), and the reward probability associated with unchosen option (side or stimulus) was also updated as follows:

$$Q_U^{RL2}(t + 1) = Q_U^{RL2}(t) - decay_{rate}(Q_U^{RL2}(t))$$
(20)

where $decay_{rate}$ is the decay (or discount) rate of the value of the unchosen option. In RL2, $DV_i = Q_i^{RL2}$.

*Loss-memory component.* The loss-memory component influences stay/switch strategy in response to receiving no reward. In unrewarded trials, the value of the loss-memory component for the chosen option ($L_C(t + 1)$) is the negative expected reward prediction error, and in rewarded trials, the value of the component is 0:

$$L_C(t + 1) = \begin{cases} 0 & if\ R(t) = 1 \\ -E_{rpe}(t + 1) & if\ R(t) = 0 \end{cases}$$
(21)

where $E_{rpe}$ denotes the expected unsigned reward prediction error.

The expected unsigned reward prediction error tracks expected uncertainty and is updated on every trial as follows:

$$E_{rpe}(t + 1) = E_{rpe}(t) + \gamma(|R(t) - Q_C^{RL2}(t)| - E_{rpe}(t)) \quad (22)$$

where $\gamma$ is the decay rate for expected reward prediction error and $R(t) - Q_C^{RL2}(t)$ is the reward prediction error on the current trial. Because the value of the loss-memory component is proportional to expected uncertainty, the no reward outcome has a greater influence on choice during times of high uncertainty.

*Choice-memory component.* The choice-memory component influences stay/switch strategy in response to selection of the better/worse option and is already known to be important for explaining behavior in mice and monkeys[8,15,38]. The values of choice memory for the chosen option (side or stimulus), $C_C$, and for the unchosen option (side or stimulus), $C_U$, are updated as follows:

$$C_C(t + 1) = C_C(t) + \gamma(1 - C_C(t))$$

$$C_U(t + 1) = C_U(t) - \gamma(C_U(t)) \quad (23)$$

where $\gamma$ represents the decay rate for the choice value.

*Models with loss- and/or choice-memory components.* The loss-memory and choice-memory components are weighted with fitted parameters and summed with learned reward values to determine the decision values for different models. Below we use the notation RL1 and RL2 to denote that the standard reward values, $Q_C(t)$ and $Q_U(t)$, that are updated based on the update rules of RL1 and RL2, respectively.

In the full model (RL2 + CM + LM), the decision values related to the chosen and unchosen options in trial t, $DV_C(t + 1)$ and $DV_U(t + 1)$, are computed as follows:

$$DV_C(t + 1) = Q_C^{RL2}(t + 1) + \omega_{LM} \times L_C(t + 1) + \omega_{CM} \times C_C(t + 1)$$

$$DV_U(t + 1) = Q_U^{RL2}(t + 1) + \omega_{CM} \times C_U(t + 1) \quad (24)$$

where $\omega_{LM}$ and $\omega_{CM}$ are free parameters that determine the relative weight of the loss-memory and choice-memory components, respectively.

In the full model, the same $\gamma$ was used for both the choice- and loss-memory components because we found that a model with different $\gamma$ fitted for the two components fit worse (based on AIC) than a model with one $\gamma$ shared between the components for mice and monkeys.

In RL2 + LM, the decision values are computed as follows:

$$DV_C(t + 1) = Q_C^{RL2}(t + 1) + \omega_{LM} \times L_C(t + 1)$$

$$DV_U(t + 1) = Q_U^{RL2}(t + 1) \quad (25)$$

In RL2 + CM, the decision values are computed as follows:

$$DV_C(t + 1) = Q_C^{RL2}(t + 1) + \omega_{CM} C_C(t + 1)$$

$$DV_U(t + 1) = Q_U^{RL2}(t + 1) + \omega_{CM} C_U(t + 1) \quad (26)$$

In RL1 + CM, the decision values are computed as follows:

$$DV_C(t + 1) = Q_C^{RL1}(t + 1) + \omega_{CM} C_C(t + 1)$$

$$DV_U(t + 1) = Q_U^{RL1}(t + 1) + \omega_{CM} C_U(t + 1) \quad (27)$$

In all models except the multiple timescales model, the probability of selecting the left side (or circle stimulus) is represented as a sigmoid function of the difference in estimated reward probabilities or values for the left and right sides (respectively, circle and square stimuli). Hence, the estimated probability of choosing the left side for mice (or circle for monkeys) in trial t, $P_{Left(Circle)}(t)$, is equal to:

$$P_{Left(Circle)}(t + 1) = \left(1 + e^{-\beta*(DV_{Left(Circle)}(t\,1) - DV_{(Right)(Square)}(t+1))}\right)^{-1} \quad (28)$$

where $\beta$ is the inverse temperature (or stochasticity in choice) that quantifies sensitivity of choice to the difference in decision values.

Values of $decay_{rate}$, $\gamma$, $\alpha_{rew}$, and $\alpha_{unrew}$ ranged from 0 to 1 for all models, and values of $\beta$ ranged from 0 to 100. For fit of mouse data, $\gamma$ was fit as a free parameter, but for fit of monkey data, $\gamma$ was fixed as $\gamma = mean(\alpha_{rew}, \alpha_{unrew})$ such that learning in choice- and loss-memory components occurred at the same rate as the acquisition of reward values. This was done because models with fixed $\gamma$ had lower mean AIC than models with fitted $\gamma$ for monkey data and models with fixed $\gamma$ had higher mean AIC than models with fitted $\gamma$ for mouse data (mean AIC; mice data: fixed $\gamma$: $AIC_{RL2+CM} = 385.65, AIC_{RL2+LM} = 376.26, AIC_{RL2+CM+LM} = 374.65$; fitted $\gamma$: $AIC_{RL2+CM} = 381.70, AIC_{RL2+LM} = 376.47, AIC_{RL2+CM+LM} = 372.97$; monkey data: fixed $\gamma$: $AIC_{RL2+CM} = 43.10, AIC_{RL2+LM} = 44.13, AIC_{RL2+CM+LM} = 43.64$; fitted $\gamma$: $AIC_{RL2+CM} = 44.16, AIC_{RL2+LM} = 45.22, AIC_{RL2+CM+LM} = 44.32$). This difference may be attributable to different task structure: a superblock for monkeys is only 80 trials, whereas a session for mice is much

longer, making the threshold for how useful a parameter must be on a trial-by-trial basis to be added to a model more stringent for monkeys.

In the above models, values of $\omega_{LM}$ and $\omega_{CM}$ varied from −1 to 1, such that the effects of recent loss and choice on future choice could increase either staying or switching behavior. To test the effects of negative choice-memory weights, we also two additional models, RL2 + LM+ and RL2 + CM+. In RL2 + LM+ and RL2 + CM+, the decision values are computed as in Eqs. (24), (25), respectively, however, $\omega_{LM}$ and $\omega_{CM}$ only range from 0 to 1 instead of −1 to 1.

*Multiple timescales model.* We also fit and simulated one additional model based on learning across multiple timescales (Iigaya et al.[31]). In this model, the values for options are updated across three timescales, $\tau_{fast-1} = 2, \tau_{fast-2} = 20, \tau_{slow} = 100$ trials. The reward values for the chosen and unchosen options computed on timescale $\tau_i$, $(Q_{C,\tau_i}^{Time}(t)$ and $Q_{U,\tau_i}^{Time}(t))$ are updated as follows:

$$Q_{C,\tau_i}^{Time}(t + 1) = \left(1 - \frac{1}{\tau_i}\right)Q_{C,\tau_i}^{Time}(t) + \left(\frac{1}{\tau_i}\right)R(t)$$

$$Q_{U,\tau_i}^{Time}(t + 1) = \left(1 - \frac{1}{\tau_i}\right)Q_{U,\tau_i}^{Time}(t) \quad (29)$$

which is equivalent to the RL2 update rule with $\alpha_{rew} = \alpha_{unrew} = decay_{rate} = 1/\tau_i$.

The decision value for the chosen (unchosen) option $(DV_{C(U)}(t))$ is then a weighted sum of the three reward values computed on different timescales:

$$DV_{C(U)}(t + 1) = \omega_{fast-1} * Q_{C(U),\tau_{fast-1}}^{Time}(t + 1) + \omega_{fast-2} * Q_{C(U),\tau_{fast-2}}^{Time}(t + 1) + \omega_{slow} * Q_{C(U),\tau_{slow}}^{Time}(t + 1) \quad (30)$$

where $\omega_{fast-1}, \omega_{fast-2}$, and $\omega_{slow}$ are fitted parameters that range from 0 to 1 and determine the contribution of different timescales to decision making. $\omega_{fast-1}, \omega_{fast-2}$, and $\omega_{slow}$ are normalized such that they sum to 1.

Finally, the probability of choosing the left side (circle stimulus) is computed as follows:

$$P_{Left(Circle)}(t + 1) = \frac{DV_{Left(Circle)}(t + 1)}{DV_{Left(Circle)}(t + 1) + DV_{Right(Square)}(t + 1)} \quad (31)$$

We also tested a few modified versions of the timescale model that incorporated fitting a beta parameter, using a sigmoid decision rule, fitting instead of fixing the $\tau$ parameters, and integrating learning on multiple timescales with RL2. However, none of modified timescale models fit or captured metrics better than RL2 + CM + LM for mice or RL2 + CM for monkeys, so we only present the original multiple timescales model.

**Model fitting and simulations.** We used the standard maximum likelihood estimation method to fit and estimate the best-fit parameters for the models described above. One set of model parameters was fit to each session of mouse data and each superblock of monkey data. We then used estimated parameters across sessions (in mice) and superblocks (in monkeys) to generate the distributions of parameters for each model (Supplementary Fig. 8). When fitting and simulating RL models with mouse data, we treated miss and no-go trials as if they had not occurred.

To quantify goodness-of-fit, we computed the Akaike Information Criterion (AIC) for each session (for mouse data) or superblock (for monkey data):

$$AIC = -2 \times log\text{-}likelihood + 2p \quad (32)$$

where $p$ is the number of free parameters in a given model. To test for significant differences in AIC, we conducted paired samples $t$ tests comparing the mean of AIC of each model with the mean AIC of the best-fitting model (Supplementary Table 2).

To compute the probability that a given model is the best model given the data and set of candidate models, we used AIC values to compute the Akaike weights[56,57] for the $i$th model $(M_i)$ in a set of $k$ models, $\{M_1, M_2, \ldots, M_k\}$, as follows:

$$\Delta A\bar{I}C(M_i) = A\bar{I}C(M_i) - \min(\{A\bar{I}C(M_1), A\bar{I}C(M_2), \ldots, A\bar{I}C(M_k)\})$$

$$w_i = \frac{e^{-\frac{1}{2}*\Delta A\bar{I}C(M_i)}}{\sum_{j=1}^{k} e^{-\frac{1}{2}*\Delta A\bar{I}C(M_j)}} \quad (33)$$

where $A\bar{I}C(M_i)$ indicates the mean AIC for $M_i$, $\Delta A\bar{I}C(M_i)$ is the difference between the mean AIC for $M_i$ and the minimum mean AIC out of the set of candidate models, and $w_i$ indicates the Akaike weight for $M_i$.

To quantify an absolute measure of goodness-of-fit, we also computed the McFadden R[2][58] for each model:

$$McFadden\,R^2 = 1 - \frac{\sum_{sessions} log\text{-}likelihood}{\sum_{sessions} log\text{-}likelihood_{Null}} = 1 - \frac{\sum_{sessions} log\text{-}likelihood}{\sum_{sessions} n \times \ln(0.5)} \quad (34)$$

where $n$ is the number of trials in a given session or superblock.

One hundred model simulations were performed per session using best-fit parameters. The large number of simulations allowed us to estimate the population distributions of all metrics. Finally, we conducted additional simulations of RL2 using random parameter values to examine the relationship between parameters and entropy-based metrics. For these simulations, $\alpha_{rew}$, $\alpha_{unrew}$, and $\beta$ varied in the range of $(0,\infty)$ and $decay_{rate}$ was set to 0.1.

**Data analyses and stepwise regressions.** Stepwise regressions were conducted using MATLAB's (R2019a) stepwiselm and stepwisefit functions. The criterion for adding or removing terms from the model was based on an $F$-test of the difference in sum of squared error resulting from the addition or removal of a term from the model. A predictor was added to the model if the $p$ value of the $F$-test was <0.0001, and a predictor was removed from the model if the $p$ value of the $F$-test was >0.00011.

We note that there were fewer blocks used in the full model stepwise regression because some of the specific entropy-based metrics were not defined for certain blocks, e.g., if a mouse or monkey never won on the worse option (worse side or stimulus) in a block, then $ERODS_{W+}$ was undefined for that block. This resulted in the exclusion of around 500 blocks for mice and 700 blocks for monkeys in the final regression.

We also conducted tenfold cross-validated regressions to predict deviation from matching (Supplementary Fig. 6) using MATLAB's (R2019a) fitrlinear and kfoldPredict functions. More specifically, stepwise regressions were performed on a set of possible predictors to determine which predictors to include in the final regression model. Then, cross-validated regressions were computed to predict deviation from matching using the set of predictors included in the final stepwise regression model.

**Reporting summary.** Further information on experimental design is available in the Nature Research Reporting Summary linked to this paper.

## Data availability

The raw and processed data generated in this study have been deposited in a GitHub repository accessible at https://doi.org/10.5281/zenodo.5501693[59]. Mouse and monkey data analyzed here have been analyzed previously; see[9] for previous analysis of mouse data, and see[35,54] for previous analyses of monkey data. Source data are provided as a Source Data file.

## Code availability

The code used for calculation of all behavioral metrics, data analyses, fitting choice data, and plotting figures is available at https://doi.org/10.5281/zenodo.5501693[58].

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

## Acknowledgements
This work is supported by the National Institutes of Health (Grants R01DA047870 to A.S. and R01NS104834 to J.C.).

## Author contributions
A.S., M.S., and E.T. conceived and designed the study; B.B., J.C., and V.C. conceived and designed the experiments; B.B. and V.C. collected the experimental data; A.S., M.S., and E.T. developed metrics and computational models; M.S. and E.T. implemented algorithms and models and analyzed the experimental and simulated data; E.T. prepared the figures; A.S. and E.T. wrote the first draft; All authors contributed to revising and editing the paper.

## Competing interests
The authors declare no competing interests.
