## [Peer Review File · Nature Communications]

Entropy-based metrics for predicting choice behavior based on local response to rewardREVIEWER COMMENTS

Reviewer #1 (Remarks to the Author):

Trepka and colleagues develop novel entropy-based measures of choice behavior and use these to study the behavior of mice and monkeys performing different repeated choice tasks under stochastically changing reinforcement contingencies. They use these entropy-based metrics along with other classic model-independent measures to determine which are most associated with undermatching, a behavior where animals allocate fewer than expected choices to the better option leading to suboptimal reward intake. They show that their entropy-based metrics explain variability in undermatching, and show specifically that the strongest correlation is with the entropy metric capturing the consistency of choice strategy after not receiving a reward from choosing the worst option. The authors then use this insight to extend common reinforcement learning (RL) models to better account for trial-by-trial choice behavior. I found the development of the entropy metrics clear, and useful since they are based on easily understood unconditional and conditional probabilities. These measures could be incorporated into future studies as useful model-independent measures. I found the logic of the the RL-based part of the paper sensible, complementing the model-independent measures. That said, I have major concerns that should be resolved before judging the acceptability of the paper for Nature Communications.

I was surprised when the authors noted that prior "outcome models were not examined for their ability to capture deviation from matching" (pg. 4) when the reference cited in the immediately preceding sentence is in fact focused on understanding deviations from matching using a number of model-based and model-independent methods (Iigaya et al., 2019). Indeed, Iigaya et al. contains figures illustrating several metrics plotted against the degree of behaviorally observed undermatching, so I find quite misleading the current authors' claim that "none of those studies aimed to predicted observed undermatching" (pg. 22). I think the authors' work stands quite well on its own without underselling prior work, and provides opportunity for a nuanced discussion (and even more fruitfully a model comparison) given the similarity in approach (deriving local models that explain global undermatching aided by model-independent metrics) and behavioral task (binary choice probabilistic reinforcement with stochastically varying reinforcement schedules).

The presentation of the RL model results is confusing. In part, this is due to mixing mouse and monkey results together. It's clear from the preceding model-independent metrics that the inter-species behavior is different, which is probably expected given the numerous task difference. But as I'm reading the text on pg. 20, I can't help but keep wondering why the authors keep referring to RL2 and Dyn LCM when the RL2+CM model is the winning model by AIC for monkey data? This paragraph is then followed by a breakdown of CM and LM components as if they are both novel, but the CM component is already well-known to be important for explaining choice behavior in mice, pigeons, monkeys and humans. It seems worth considering presenting the observation that previous RL models work for monkeys, but still (to be verified as in Figure 5f using RL2+CM) fail to capture the distribution of ERODS_{W-} values, but that for mice, it is different and the loss-memory component (but not the choice-memory component) is crucial. This latter point is interesting in light of prior work (e.g., Fonseca et al 2015) showing that choice-history effects in mice can be quite strong.

This leads to the other reason the model results are confusing, which is that despite undermatching being correlated with receiving no rewards following choices of the worst option in both species, a loss-memory component is only needed for the mouse data. The authors don't discuss this, but it could be due to many factors, even unrelated to species differences (overall reward rates, object versus spatial actions, baiting of rewards on unchosen option for mice which makes a certain degree of switching sensible, etc). The model fits are also worse for the mouse data (R^2 roughly half of that for the monkey), which may also merit discussion. Although the authors argue that their approach is a general framework for explaining global behavior, the inclusion of two datasets with marked differences warrants discussion about how application of the framework yielded different conclusions across datasets.

Overall, the exposition of the reinforcement learning models is not clear. This seems to stem from not explicitly mentioning that the models are all nested, some confusing/inconsistent notation, and some unjustified assumptions. For example, for the sake of clarity, it seems like the full model should be referred to as RL2+CM+LM in keeping with the logic for the other model names rather than Dyn. LCM. The symbol $W^{\text{RL2}}_C(t)$ in eq. 24 is undefined, as is $V^{\text{CM}}_C(t)$ in eq. 25. ω_1 and ω_2 are used in eq. 25, but I assume these refer to ω_{LM} and ω_{CM} , respectively. Even more confusing, I think ω_{LM} is referred to as ω_{reward} in the supplementary figures? If I understand correctly, both the loss-memory and choice-memory component influence subsequent choice, but do not get incorporated into subsequent expected value for each option. If so, the notation in eq. 25 does not distinguish this, and one would assume from the notation in eq. 18-19 that these components do indeed contribute to the updated values (indeed they are all symbolized by V , presumably for value). These could be more clearly indicated as choice biases by putting them in eq. 22. Finally, why constrain γ in all of the RL2+ models to equal the mean(α_{rew} , α_{unrew})? Was this tested? If so, it should be stated, otherwise it's not clear why this parameter should be restricted this way. Together, these make understanding the relationships between the models extremely difficult, and I would recommend rewriting the methods and results surrounding these models, with an emphasis on explaining the novel component (loss-memory), as the choice-memory component is commonly incorporated in RL models.

The authors describe perfect matching as optimal in the mouse task. This is true for a stationary environment with fixed reward probabilities (Houston & McNamara 1981), but this is not generally true in a stochastic environment where reward probabilities change unbeknownst to the subjects (Iigaya et al. 2019).

Minor concerns

I'm confused about how the RL models were actually fit to the data. The authors state that they used maximum-likelihood, but do not explicitly state that a single set of parameters (for each model) was fit to the totality of the data from each species. Is this the case? I guessed so since there are only point estimates of AIC values in the tables and Figures. However, I was confused by Figure S7, where parameter distributions are shown. How were these calculated? Please clarify in the methods.

Miss trials are excluded for mice, but please report the number of of these trials that were excluded. Also in computing behavioral metrics based on the action or reward on the last trial, how was this handled for trials preceded by a miss or nogo trial. Were these dropped from the calculations? How were these handled in the RL models?

Figure 1d. Legend states that c-d represent individual sessions, but the monkey data only shows a "superblock", which is only part of a session. Would be nice to see the whole session, or at least as much data as the mouse (i.e., more transitions).

Figure 3. Some transparency of the points would help readers see better whether there is mouse data underneath the monkey data.

Figure 5. A relative AIC score might be more useful here. Something like the AIC difference relative to the minimum AIC, or even better Akaike weights (Burnham & Anderson 2002) giving the relative likelihood of each model. Also, it's not clear what the utility of panels c&d are given that the distributions and cumulative distributions are plotted in e&f?

Please incorporate p-values for the Kolmogorov-Smirnov tests. I think that, unlike t-tests, most people won't be able to figure out what the D-statistic means.

Reviewer #2 (Remarks to the Author):

This paper presents a new analysis of matching behavior in both mice and monkeys. There are two main contributions. One is a new set of information-theoretic metrics for characterizing behavioral patterns. These metrics allow the authors to both capture a significant amount of variance in behavior and to identify shortcomings of existing models. The second main contribution is the identification of a new model that does a significantly better job at capturing the new metrics.

Overall, I think this is a nice example of how careful analysis of behavior can guide model development. I'm also very glad that the authors have introduced concrete alternatives to win-stay/lose-shift, which in my experience has been pretty uninformative about underlying mechanisms.

My comments are fairly minor.

1) I wasn't sure if the regression analyses with the metrics was really necessary to report in the main text. As I understand it, these analysis basically demonstrate the the metrics are capturing a substantial amount of variance in the behavior. But I think this could just be mentioned in passing, since the main goal is to use the metrics not to explain behavior directly but to guide model development.

2) Figure 4 is nearly unreadable. There's just too much going on there. Maybe it could go in the supplement and be replaced with some condensed form that is more manageable.

3) In the Discussion, the authors address how negative choice weights can explain various phenomena in the decision making literature. I want to raise two issues here. First, there is more probability mass on positive choice weights in Fig S7i. Second, I think it's somewhat problematic to assume that some phenomena are explained by positive choice weights and some by negative choice weights, consistent with many other past studies. Presumably this is not an idiosyncratic property of the subjects in these studies; possibly it could be a property of the tasks. However, this parameter is not task-dependent in the model. So the model doesn't really offer a coherent explanation of why choice weights might be negative vs. positive. They are simply free parameters. I'm not arguing that the authors have to fill this gap in the current paper, just that they need to be somewhat careful in how they talk about what their model can or cannot explain.

4) This is very minor, but I don't see why you need to define the conditional entropy in terms of the mutual information in Eq. 7. You can just define it directly.

Response to the reviewers' comments, and summary of changes made in response to the comments of the reviewers.

Title: Novel entropy-based metrics for predicting choice behavior based on local response to reward

Authors: Trepka, Spitmaan, Bari, Costa, Cohen, and Soltani

We are very thankful to all the reviewers for their careful reading of our manuscript and their thoughtful comments, suggestions, and feedback. We have performed additional analyses and made substantial changes in the revised manuscript to fully address all of the reviewers' concerns and suggestions. Below, we provide a point-by-point response to each of the concerns detailed. The corresponding changes in the revised manuscript have been clearly marked in blue and indexed by [RX.Y] (e.g., [R1.1] indicates changes in response to point 1 of Reviewer # 1, etc.).

Reviewer #1

"Trepka and colleagues develop novel entropy-based measures of choice behavior and use these to study the behavior of mice and monkeys performing different repeated choice tasks under stochastically changing reinforcement contingencies. They use these entropy-based metrics along with other classic model-independent measures to determine which are most associated with undermatching, a behavior where animals allocate fewer than expected choices to the better option leading to suboptimal reward intake. They show that their entropy-based metrics explain variability in undermatching, and show specifically that the strongest correlation is with the entropy metric capturing the consistency of choice strategy after not receiving a reward from choosing the worst option. The authors then use this insight to extend common reinforcement learning (RL) models to better account for trial-by-trial choice behavior. I found the development of the entropy metrics clear, and useful since they are based on easily understood unconditional and conditional probabilities. These measures could be incorporated into future studies as useful model-independent measures. I found the logic of the the RL-based part of the paper sensible, complementing the model-independent measures. That said, I have major concerns that should be resolved before judging the acceptability of the paper for Nature Communications."

Response: We thank the reviewer for the thoughtful and overall positive evaluation and summary of our work. We hope our answers here and corresponding changes in the revised manuscript address all of the reviewer's concerns.

“I was surprised when the authors noted that prior “outcome models were not examined for their ability to capture deviation from matching” (pg. 4) when the reference cited in the immediately preceding sentence is in fact focused on understanding deviations from matching using a number of model-based and model-independent methods (Iigaya et al., 2019). Indeed, Iigaya et al. contains figures illustrating several metrics plotted against the degree of behaviorally observed undermatching, so I find quite misleading the current authors’ claim that “none of those studies aimed to predicted observed undermatching” (pg. 22). I think the authors’ work stands quite well on its own without underselling prior work, and provides opportunity for a nuanced discussion (and even more fruitfully a model comparison) given the similarity in approach (deriving local models that explain global undermatching aided by model-independent metrics) and behavioral task (binary choice probabilistic reinforcement with stochastically varying reinforcement schedules).”

Response: We thank the reviewer for this comment and would like to point out upfront that by no means we intended to undersell previous work. The reviewer is correct that Iigaya and colleagues (2019) fit a new model to choice data during a similar task and examined correlations between an estimated model parameter and observed undermatching (Fig. 5 in Iigaya et al.). However, they did not perform model comparison nor did they show that simulated data using their model could replicate the observed variance in undermatching. Moreover, they measured undermatching over each session (and not each block) of the experiment (please see our response to [R1.12] later in the rebuttal for more discussion of this). Nonetheless, as suggested by the reviewer, we now have revised the sentence on page 4 (see [R1.1] in the revised manuscript) to focus on the opportunity for model comparison:

(page 4) “Although these models all provide compelling explanations of the emergence of matching behavior, it remains unclear how they compare in terms of fitting local choice behavior and the extent to which they replicate observed variability in matching behavior [R1.1].”

We also agree with the reviewer that the sentence on page 22 was misleading given that Iigaya et al. explained undermatching using both a model-independent metric and the values of fitted model parameters. We have removed that sentence, and now discuss the relationship between our behavioral results and those of Iigaya et al. elsewhere in the Discussion (please see our response to [R1.12] later in the rebuttal).

In addition, to further compare the ability of different models to capture choice behavior, we now have added fitting results based on the model proposed by Iigaya et al. (which we refer to as the multiple timescales model) to our model comparison. We found that the multiple timescales model was worse than RL2 (and thus our full model) in fitting choice behavior in terms of AIC for both mouse and monkey data (see **Table 1** below). Moreover, the multiple timescales model was worse than RL2 in terms of capturing the observed distributions of deviation from matching and ERODS_{w-} (see **Table 1** below). We also went further and explored alternative versions of the multiple timescales model that were more similar to RL2 including: (a) a multiple timescales model augmented with a sigmoid decision rule; (b) a multiple timescales model with fitted instead of fixed timescales; and (c) a version of RL2 with learning on two different timescales. None of these models achieved better fit or capture of metrics than our best fitting model, however, version (c) improved upon RL2 in fitting choice behavior and capturing metrics for mouse data. To avoid confusion, we only present the results from the original multiple timescales model of Iigaya et al (see [R1.1]).

Model	Model Description	Parameters	AIC	D ERODS _{w-}	D matching
Multiple Timescales	Learning on multiple timescales	$\omega_{\text{fast-1}}$,  $\omega_{\text{fast-2}}$,  ω_{slow}	402.99 (30.02*)	0.188	0.165
			49.76 (6.66*)	0.127	0.164
RL2	Income-based RL	α_{rew} , α_{unrew} , β , decay _{rate}	387.22 (14.25*)	0.121	0.091
			44.18 (1.08*)	0.072	0.101

Table 1. Comparison of fit of choice data and capture of metrics in simulations for the multiple timescales model from Iigaya et al. and RL2. Each row provides a short description of a given model, its parameters, goodness-of-fit based on the AIC, and D-values based on Kolmogorov-Smirnov tests comparing distributions of ERODS_{w-} and deviation from matching. Rows in orange and cyan correspond to mouse and monkey data, respectively [R1.1].

These results now have been included in the revised manuscript (see [R1.1]) and **Table 1** above has been incorporated in Supplemental **Table S2**. We also have added the following paragraph to the Discussion (see [R1.1]) detailing these findings and their implications:

(page 26) *“Moreover, we found that nearly all other models described here better accounted for local and global choice behavior than the multiple timescales model proposed in Iigaya et al. (2019) study. Nonetheless, it is possible that more complex models based on learning on multiple timescales may fit choice behavior better [R1.1].”*

“The presentation of the RL model results is confusing. In part, this is due to mixing mouse and monkey results together. It’s clear from the preceding model-independent metrics that the inter-species behavior is different, which is probably expected given the numerous task difference. But as I’m reading the text on pg. 20, I can’t help but keep wondering why the authors keep referring to RL2 and Dyn LCM when the RL2+CM model is the winning model by AIC for monkey data? This paragraph is then followed by a breakdown of CM and LM components as if they are both novel, but the CM component is already well-known to be important for explaining choice behavior in mice, pigeons, monkeys and humans. It seems worth considering presenting the observation that previous RL models work for monkeys, but still (to be verified as in Figure 5f using RL2+CM) fail to capture the distribution of ERODS_{W-} values, but that for mice, it is different and the loss-memory component (but not the choice-memory component) is crucial. This latter point is interesting in light of prior work (e.g., Fonseca et al 2015) showing that choice-history effects in mice can be quite strong.”

Response: We thank the reviewer for pointing out this issue and their helpful suggestion. As suggested by the reviewer, we now have revised the manuscript to discuss most of the results for the two species in separate paragraphs (see [R1.2]). The reviewer is correct that for the monkey data, the RL2+CM model actually captures the distributions of ERODS_W- and undermatching as well as the full model. As a result, we now focus our description of the monkey fitting results exclusively on the RL2+CM model and the mouse fitting results exclusively on the RL2+CM+LM model.

The main model fitting results for monkey data now read as follows (see [R1.2]):

(pages 17-18) *“In monkeys, we found that the RL2 model augmented with a CM component, which we refer to as the RL2+CM model, fit choice behavior better than RL1, RL2, and RL1+CM as indicated by lower AIC (Fig. 5b; Table S2). Although the improvement in fit of choice behavior for RL2+CM over RL2 was statistically significant (paired-samples t-test of AICs: $p =$*

$1.04 * 10^{-23}$; **Table S2**), the RL2+CM model was only twice as likely as RL2 to be the best model based on a comparison of Akaike weights.

Importantly, the RL2+CM model improved capture of the observed distribution of $ERODS_w$ in monkeys (**Fig. 5d**; two-sided Kolmogorov-Smirnov test; $D = 0.037$, $p = 8.91 * 10^{-3}$). This improvement in capturing $ERODS_w$ corresponded with similar improvements in capturing deviation from matching. The predicted distribution of deviation from matching from the RL2+CM model better replicated the observed distribution of deviation from matching than the predicted distribution from RL2 (**Fig. 5f**; two-sided Kolmogorov-Smirnov test; $D = 0.065$, $p = 2.07 * 10^{-8}$). This improvement was significant; there was an over 30% reduction in the maximum difference between CDFs in the RL2+CM model from the RL2 model [R1.2].”

The main model fitting results for mouse data now read as follows (see [R1.2]):

(page 20) “In mice, the RL2+CM+LM model (RL2 augmented with a choice-memory and a loss-memory component) fit choice behavior better than all previous RL models as indicated by a lower AIC (**Fig. 5a**). The Akaike weight for the RL2+CM+LM model was 0.84, which suggests there is a high probability that the RL2+CM+LM model is the best model out of all models examined. The RL2+CM+LM model also captured the observed distribution of $ERODS_w$ for mice better than RL2 (**Fig 5c–d**; two-sided Kolmogorov-Smirnov test; $D = 0.049$, $p = 3.77 * 10^{-7}$). Moreover, the predicted distribution of deviation from matching from the RL2+CM+LM model better replicated the observed distribution of deviation from matching than the predicted distribution from RL2 (**Fig 5e**; two-sided Kolmogorov-Smirnov test; mice: $D = 0.065$, $p = 2.19 * 10^{-12}$). This improvement corresponds to an over 20% reduction in the maximum difference between cumulative distribution functions (CDFs) for deviation from matching computed from observed and simulated data [R1.2].”

We also now have better clarified that the choice-memory component is not novel and discuss its relationship to the entropy-based metrics (see [R1.2]). To that end, the revised description of the choice-memory component now begins with the following:

(page 17) “To improve capture of option-dependent strategy, we added a common choice-memory component to estimate the effects of previous choices on subsequent decisions (Lau and Glimcher, 2005; Fonseca et al., 2015; Wittmann et al., 2020). The choice-memory component encourages either staying on or switching from options that have been chosen recently. Because

standard RL models typically choose the option with a higher value, the choice-memory component can capture strategy in response to selection of the better or worse option reflected in the option-dependent entropy-based metrics [R1.2].”

Finally, we also thank the reviewer for providing the relevant reference. We now address the relationship of these findings to previous work such as Fonseca et al., 2015 in the Discussion (see [R1.2]):

(page 25) “We also observed weak, positive choice-memory effects in mice such that mice tended to choose options that they had recently chosen. A previous study using a nearly identical task (reversal learning with same reward schedules (40/10) and baited rewards, but longer blocks) observed a much stronger, negative choice memory effect in mice (Fonseca et al., 2015). The reason for this difference is unclear given the similarity of the two tasks used. Consistent with prior studies of choice-history effects in monkeys (Lau and Glimcher, 2005), we identified strong, negative choice memory effects in monkeys such that the choice-memory component encouraged switching from recently chosen options. Thus, the incorporation of the negative weights was only important for capturing behavior in the monkey task and therefore could be task dependent [R1.2] [R2.3].”

“This leads to the other reason the model results are confusing, which is that despite undermatching being correlated with receiving no rewards following choices of the worst option in both species, a loss-memory component is only needed for the mouse data. The authors don’t discuss this, but it could be due to many factors, even unrelated to species differences (overall reward rates, object versus spatial actions, baiting of rewards on unchosen option for mice which makes a certain degree of switching sensible, etc).”

Response: We thank the reviewer for pointing this out and appreciate the suggested explanations. We now elaborate on potential explanations for why a loss-memory component is only needed for the mouse data in the Discussion as follows (see [R1.3]):

(page 24) “Despite the significant correlation between ERODSW- and deviation from matching in both species, the novel loss-memory component introduced here only improved fit of choice behavior and capture of metrics in the full model in mice. This finding may be related to the close correspondence between reward- and option-dependent strategies in the monkey task since

winning (respectively, losing) almost always corresponds with choosing the better (respectively, worse) side. Due to this significant overlap, one component may be sufficient to capture both strategies. In the mouse task, however, these strategies were dissociated because losing was likely when choosing either the better or worse option (but more for the worse option). This could explain why for monkeys, the LM component improved capture of entropy-based metrics and deviation from matching in the RL2+LM model relative to the RL2 model but was not useful in conjunction with the choice-memory component. Moreover, we observed a higher overall probability of switching in mice than in monkeys, indicating that mice occasionally switch from the more-rewarding side to harvest baited rewards on the less-rewarding side, whereas monkeys typically exploit the more-rewarding stimulus. Because of this, a loss-memory component that encourages switching in response to loss would be more helpful in capturing that behavior in mice than in monkeys [R1.3]."

"The model fits are also worse for the mouse data (R^2 roughly half of that for the monkey), which may also merit discussion."

Response: We thank the reviewer for pointing this out and have added the following explanation to the Discussion to address it (see [R1.4]):

(page 26) "The model fits were also worse for mouse data than the monkey data in terms of explained variance in choice behavior, likely due to differences in the overall entropy in choice behavior and task structure. More specifically, mice showed higher average entropy in their choice behavior than monkeys across different measures, suggesting that the observed difference in the quality of fit occurred because mice choice behavior was more random and thus harder to predict. In addition, sessions in the mouse task were longer than superblocs in the monkey task, so the same number of parameters were used to account for more choices in mice than in monkeys, resulting in an overall poorer fitting quality [R1.4]."

"Although the authors argue that their approach is a general framework for explaining global behavior, the inclusion of two datasets with marked differences warrants discussion about how application of the framework yielded different conclusions across datasets."

Response: We thank the reviewer for raising this concern. Our intention was not to claim that entropy-based metrics can fully explain global behavior. Instead, we meant to

say that they can be used to identify limitations of current models and construct better models. We have addressed issues related to differences in the two datasets in our three previous responses above. In addition, we now have clarified our claims about the generality and use of our approach and detailed its limitations in the Discussion more clearly (see [R1.5]). More specifically, we have removed the claim that our approach is a general framework for *explaining* global behavior and replaced it with the following more conservative statement about *exploring* choice behavior:

(page 23) *“Our aim here was not to find the best model for capturing all aspects of behavior but instead to provide a framework for how local response to reinforcement can be used to guide model development and explore interesting properties of local and global choice behavior [R1.5].”*

In addition, we have also added the following paragraph to the Discussion regarding the limitations of our approach:

(page 24) *“Although aforementioned differences in results for these two datasets may be partially explained by differences in task structure and species, they also highlight the limitations of using entropy-based metrics to guide model development. Entropy-based metrics describe properties of choice behavior that are helpful for making educated guesses about model structure, but alone, cannot provide a generative account of behavior [R1.5] [R2.3].”*

“Overall, the exposition of the reinforcement learning models is not clear. This seems to stem from not explicitly mentioning that the models are all nested, some confusing/inconsistent notation, and some unjustified assumptions.”

Response: We thank the reviewer for pointing out the lack of clarity in our explanation and presentation of the RL models. To address this, we now have completely rewritten the Methods section and corrected some of the notational and structural inconsistencies throughout the manuscript as explained below (see [R1.6] in the revised manuscript). We also now have explicitly stated that models are all nested as follows:

(page 33) *“Models were defined in a nested fashion with subsequent models building on the update rules of their predecessor [R1.6].”*

“For example, for the sake of clarity, it seems like the full model should be referred to as

RL2+CM+LM in keeping with the logic for the other model names rather than Dyn. LCM. The symbol $W^{\{RL2\}}_C(t)$ in eq. 24 is undefined, as is $V^{\{CM\}}_C(t)$ in eq. 25.”

Response: We thank the reviewer for pointing out these inconsistencies and their helpful suggestion. We now refer to the full model as RL2+CM+LM throughout the paper. We have also updated those equations in the revised Methods section (see [R1.7]) as to ensure all variables are completely defined (the previous Eq. 24 is now Eq. 22; previous Eq. 25 is now Eq. 24).

“ ω_1 and ω_2 are used in eq. 25, but I assume these refer to $\omega_{\{LM\}}$ and $\omega_{\{CM\}}$, respectively. Even more confusing, I think $\omega_{\{LM\}}$ is referred to as $\omega_{\{reward\}}$ in the supplementary figures?”

Response: Yes, the reviewer is correct. We now refer to these variables as ω_{CM} and ω_{LM} throughout the methods and figures (see [R1.8]).

“If I understand correctly, both the loss-memory and choice-memory component influence subsequent choice, but do not get incorporated into subsequent expected value for each option. If so, the notation in eq. 25 does not distinguish this, and one would assume from the notation in eq. 18-19 that these components do indeed contribute to the updated values (indeed they are all symbolized by V , presumably for value). These could be more clearly indicated as choice biases by putting them in eq. 22.”

Response: We thank the reviewer for this suggestion and apologize for the confusing notation. The reviewer’s understanding was correct; the loss-memory and choice-memory components influence choice, but not the update of expected reward value for each option. In the revised Methods section on models, we now have implemented the suggested changes and now use Q to symbolize reward values that are updated and DV to symbolize the decision values that are input into the sigmoid function as described below (see [R1.9]):

(page 33) *“In all models except the multiple timescales model, reward values associated with the right and left sides (Q_{Right} and Q_{Left}) for mice or circle and square stimuli (Q_{circle} and Q_{square}) for monkeys were updated differently depending on whether a given choice was rewarded or not. Some of the models incorporated additional loss- or choice-memory components that influenced*

choice but did not affect the update of reward values. As such, we refer to the final reward and non-reward values used for decision making as decision values, DV (e.g., DV_{circle}), to distinguish them from the updated reward values [R1.9]."

"Finally, why constrain γ in all of the RL2+ models to equal the mean(α_{rew} , α_{unrew})? Was this tested? If so, it should be stated, otherwise it's not clear why this parameter should be restricted this way."

Response: We thank the reviewer for questioning this assumption. The assumption behind the constraint was that learning from choice history occurs at a similar rate as learning from reward history. To address this issue and test this assumption explicitly, we have fit choice data using models with fixed and free γ . Indeed, we found that all RL2+ models with fixed γ fit choice data better than all RL2+ models with free γ for monkey choice data. The reverse was true for mice: all RL2+ models with free γ fit choice data significantly better than RL2+ models with fixed γ . We suspect that this occurred because a superblock for monkeys was only 80 trials, whereas a session for mouse experiment was much longer and thus, the threshold for how useful a parameter must be on a trial-by-trial basis to be added to a model is slightly more stringent for monkey data. As a result, in the revised manuscript, we now have re-fit and simulated all of the RL2+ models for mice data to incorporate a free γ parameter. We indicate this in all model figures and tables by appending (fix γ) and (fitted γ) to model names for monkeys and mice, respectively (see [R1.10]). We also have added the following paragraph to the Methods detailing this change (see [R1.10]):

(page 37) "For fit of mouse data, γ was fit as a free parameter, but for fit of monkey data, γ was fixed as $\gamma = \text{mean}(\alpha_{rew}, \alpha_{unrew})$ such that learning in choice- and loss-memory components occurred at the same rate as the acquisition of reward values. This was done because models with fixed γ had lower AIC than models with fitted γ for monkey data and models with fixed γ had higher AIC than models with fitted γ for mouse data. This difference may be attributable to different task structure: a superblock for monkeys is only 80 trials, whereas a session for mice is much longer, making the threshold for how useful a parameter must be on a trial-by-trial basis to be added to a model more stringent for monkeys [R1.6] [R1.10]."

"Together, these make understanding the relationships between the models extremely difficult,

and I would recommend rewriting the methods and results surrounding these models, with an emphasis on explaining the novel component (loss-memory), as the choice-memory component is commonly incorporated in RL models.”

Response: We appreciate this recommendation and as explained above, have substantially revised both the Methods and Results sections to emphasize the novel loss-memory component. Moreover, we now have dedicated a subsection in the Methods to explain the loss-memory component as follows (see [R1.11]):

(pages 34-35) *“The loss-memory component influences stay/switch strategy in response to receiving no reward. In unrewarded trials, the value of the loss-memory component for the chosen side ($L_C(t + 1)$) is the negative expected reward prediction error, and in rewarded trials, the value of the component is 0:*

$$L_C(t + 1) = \begin{cases} 0 & \text{if } R_c(t) = 1 \\ -E_{rpe}(t + 1) & \text{if } R_c(t) = 0 \end{cases} \quad (\text{Eq. 21})$$

where E_{rpe} denotes the expected unsigned reward prediction error.

The expected unsigned reward prediction error tracks expected uncertainty and is updated on every trial as follows:

$$E_{rpe}(t + 1) = E_{rpe}(t) + \gamma(|R_c(t) - Q_C^{RL2}(t)| - E_{rpe}(t)) \quad (\text{Eq. 22})$$

where γ is the decay rate for expected reward prediction error and $R_c(t) - Q_C^{RL2}(t)$ is the reward prediction error on the current trial [R1.7]. Because the value of the loss-memory component is proportional to expected uncertainty, the no reward outcome has a greater influence on choice during times of high uncertainty [R1.6] [R1.11].”

“The authors describe perfect matching as optimal in the mouse task. This is true for a stationary environment with fixed reward probabilities (Houston & McNamara 1981), but this is not generally true in a stochastic environment where reward probabilities change unbeknownst to the subjects (Iigaya et al. 2019).”

Response: We thank the reviewer for this comment and helpful references. We agree that descriptions of a particular strategy as optimal are questionable, especially in a stochastic environment in which performance and matching can be defined differently and across various timescales. For example, in our manuscript we measure performance as the total number of harvested rewards in “each block” of trials (with a specific

reward probabilities) and undermatching as the difference between choice and reward fractions in each block. In contrast, Iigaya et al. (2019) use “harvesting efficiency”, equal to the number of rewards harvested divided by the maximum number of rewards that could have been collected, in “each session” of experiment (consisting of multiple blocks) as a measure of performance and quantify undermatching as the difference between the slope of choice vs. reward fractions and one in each session. This could explain why Iigaya et al. find undermatching to be optimal despite “optimality of matching behavior” within a single block of their task.

Therefore, in the revised manuscript, we now have downplayed most discussions of optimality and instead, focused on explaining variability in matching which is one of the main points of our study. Specifically, all sentences relating perfect matching and optimal behavior have been removed. Moreover, in the Results and Discussion sections, we now clearly mention that our claim of undermatching being sub-optimal in the mouse task is based on the observed strong negative correlation between undermatching and the total harvested rewards (see [R1.12]). More specifically, we have revised the following text in the Results section:

(page 6) *“As a result, selecting the worse side (side with lower base reward rate) occasionally can improve the overall total harvested reward because due to baiting, the reward rate of the worse side will exceed the reward rate of the better side if the worse side is not chosen for many trials [R1.12].”*

(page 13) *“Out of all behavioral metrics tested, the probability of winning and probability of staying had the two strongest correlations with deviation from matching for mice and monkeys, respectively, but each metric individually only captured about 25% of variance in deviation from matching (Fig. 4). The correlation between the probability of winning (total harvested rewards) and deviation from matching was positive such that increased total harvested rewards corresponded with less undermatching [R1.12].”*

And we have added the following comment to the Discussion to compare our findings with those of Iigaya et al.:

(page 25) *“The goal of our approach, to predict and develop new generative models to explain undermatching, was similar to a recent study that suggested limited undermatching results in optimal performance in stochastic environments and proposed learning on multiple timescales to account for such undermatching (Iigaya et al., 2019). In contrast, we identified a positive*

correlation between reward harvesting and deviation from matching which suggests that the degree of undermatching observed here corresponded with suboptimal choice. This difference between Iigaya et al (2019) and our study could be due to differences in how performance and undermatching are defined. More specifically, here we measure performance as the total number of harvested rewards in each block of trials and undermatching as the difference between choice and reward fractions in each block. In contrast, Iigaya et al (2019) use “harvesting efficiency”, equal to the number of rewards harvested divided by the maximum number of rewards that could have been collected in each session of experiment (consisting of multiple blocks) as a measure of performance and quantify undermatching as the difference between the slope of choice vs. reward fractions and one in each session [R1.12].”

Minor concerns

“I’m confused about how the RL models were actually fit to the data. The authors state that they used maximum-likelihood, but do not explicitly state that a single set of parameters (for each model) was fit to the totality of the data from each species. Is this the case? I guessed so since there are only point estimates of AIC values in the tables and Figures. However, I was confused by Figure S7, where parameter distributions are shown. How were these calculated? Please clarify in the methods.”

Response: We thank the reviewer for mentioning these ambiguities in our presentation. Actually, one set of model parameters was fit to data from each individual sessions for mouse data and each individual superblocks for monkey data. Therefore, each session for mice and each superblock for monkeys yielded a different set of parameters that are used for plotting distributions shown in Figure S7. We now have modified the Methods section to clarify these points as follows (see [R1.13]):

(page 38) “One set of model parameters was fit to each session of mouse data and each superblock of monkey data. We then used estimated parameters across sessions (in mice) and superblocks (in monkeys) to generate the distributions of parameters for each model (Fig. S8) [R1.13].”

The AIC values reported in the tables and figures are the average AIC values across all sessions or superblocks. Because there is a different number of trials in each session for mouse data, the standard error of AIC is not very meaningful (AIC depends on the

number of trials similar to -LL) and thus, is not included. Instead, we include asterix in **Table S2** to indicates significant differences in AIC for each model from the full model using paired-samples *t*-tests. Using this approach, we found that all models were significantly worse than the best-fitting model in both mice and monkeys. We now have clarified this in the Methods section as follows (see [R1.13]):

(page 38) *“To test for significant differences in AIC, we conducted paired-samples t-tests comparing the mean of AIC of each model with the mean AIC of the best-fitting model (Table S2) [R1.13].”*

“Miss trials are excluded for mice, but please report the number of of these trials that were excluded. Also in computing behavioral metrics based on the action or reward on the last trial, how was this handled for trials preceded by a miss or nogo trial. Were these dropped from the calculations? How were these handled in the RL models?”

Response: We thank the reviewer for asking about these details that we now have included in the Methods section (see [R1.14]).

With regard to the number of excluded miss/nogo trials, we have added the following:

(page 26) *“Miss trials, in which the mouse did not make a choice, and no-go trials were excluded for all analyses described here. In total, 1706 miss trials (average of 3.64 per session) and 7893 no-go trials (average of 16.83 trials per session) were excluded from our analyses [R1.14].”*

With regard to how metrics were computed with miss/nogo trials, we have added the following:

(page 28) *“When computing metrics based on action or reward in the previous trial for mice, we treated each miss trial as though the trial did not exist. For example, if a mouse chose left and was rewarded on trial *t*, did not respond on trial *t*+1 (miss trial), then chose left on trial *t*+2, trial *t*+2 would be labeled as win-stay [R1.14].”*

With regard to how miss/nogo trials were handled in fitting RL models, we have added the following:

(page 38) “When fitting and simulating RL models with mouse data, we treated miss and no-go trials as if they had not occurred [R1.14].”

“Figure 1d. Legend states that c-d represent individual sessions, but the monkey data only shows a “superblock”, which is only part of a session. Would be nice to see the whole session, or at least as much data as the mouse (i.e., more transitions).”

Response: We thank the reviewer for this comment. We previously did not include a larger session for monkey data because in the monkey task, different colored stimuli were used in adjacent superblocks such that divisions between superblocks were not reversals, but instead new learning periods. We now have revised **Figure 1** (see Figure 1 below) to show 400 trials of a session from a monkey (same number of trials as mice) and marked divisions between superblocks with solid vertical lines to avoid any confusion (see [R1.15]).

Figure 1. Schematic of the experimental paradigms in mice and monkeys and basic behavioral results. (a–b) Timeline of a single trial during experiments in mice (a) and monkeys (b). To initiate a trial, mice received an olfactory “go” cue (or “no go” cue in 5% of trials) (a), and monkeys fixated on a central point (b). Next, animals chose (via licks for mice and saccades

for monkeys) between two options (left or right tubes for mice and circle or square for monkeys) and then received a reward (drop of water and juice for mice and monkeys, respectively) probabilistically based on their choice. (c–d) Average choice and reward using a sliding window with a length of 10 for a representative session in mice (c) and five superblocs of a representative session in monkeys (d). Vertical grey dashed lines indicate trials where reward probabilities reversed. Vertical grey solid lines indicate divisions between superblocs in the monkey experiment [R1.15]. (e–f) Average choice and reward fractions around block switches using a non-causal smoothing kernel with a length of 3 separately for all blocks with a given reward schedule in mice (e) and monkeys (f). The better (or worse) option is the better (or worse) option prior to the block switch. Trial zero is the first trial with the reversed reward probabilities. Average choice fractions for the better option (side or stimulus) are lower than average reward fractions for that option throughout the block for both mice and monkeys, corresponding to undermatching behavior.

“Figure 3. Some transparency of the points would help readers see better whether there is mouse data underneath the monkey data.”

Response: We appreciate this suggestion and have added transparency of points to revised **Figure 3** as shown in Figure 2 below (see [R1.16]).

Figure 2. Relationship between new entropy-based metrics and win-stay, lose-switch strategies. (a–c) Plotted are ERDS and ERDS decompositions as a function of $p(\text{win})$, $p(\text{lose})$, win-stay, and lose-switch. Darker colors correspond to larger values of metrics. For the plot in panel A, $p(\text{win})$ is set to 0.5. Observed entropy-based metrics and constituent probabilities for each block for mice (orange dots) and monkeys (green dots) are superimposed on surfaces. (d–f) EODS and EODS decompositions as a function of the probabilities of choosing the better and worse option, $p(\text{better})$ and $p(\text{worse})$, probability of stay on the better option, and probability of switch from the worse option. For the plot in panel (d), $p(\text{better})$ is set to 0.5. For all plots, the units of entropy-based metrics are bits. (g–i) Same as in panels (a–c) but using heatmap. (j–l) Same as in panels (d–f) but using heatmap [R1.16].

“Figure 5. A relative AIC score might be more useful here. Something like the AIC difference relative to the minimum AIC, or even better Akaike weights (Burnham & Anderson 2002) giving the relative likelihood of each model. Also, it’s not clear what the utility of panels c&d are given that the distributions and cumulative distributions are plotted in e&f?”

Response: We appreciate this suggestion and now have added Akaike weights to **Figure 5** in the main text and AIC difference in **Table S2**. As suggested, we also have removed panels c&d from the revised **Figure 5** (see [R1.17]).

“Please incorporate p-values for the Kolmogorov-Smirnov tests. I think that, unlike t-tests, most people won’t be able to figure out what the D-statistic means.”

Response: We thank the reviewer for pointing this out and now have included p-values of the Kolmogorov-Smirnov tests to the main model fitting figure. We note that these p-values are very small due to the large number of simulations performed. We also have added a text to help clarify the meaning of the D-value for more general audience as explained below (see [R1.18]):

(page 16) “To evaluate the similarity of observed and predicted distributions of entropy-based metrics and matching, we computed Kolmogorov’s D statistic that measures the maximum difference (or distance) between two empirical cumulative distribution functions [R1.18].”

Reviewer #2

“This paper presents a new analysis of matching behavior in both mice and monkeys. There are two main contributions. One is a new set of information-theoretic metrics for characterizing behavioral patterns. These metrics allow the authors to both capture a significant amount of variance in behavior and to identify shortcomings of existing models. The second main contribution is the identification of a new model that does a significantly better job at capturing the new metrics.

Overall, I think this is a nice example of how careful analysis of behavior can guide model development. I’m also very glad that the authors have introduced concrete alternatives to win-stay/lose-shift, which in my experience has been pretty uninformative about underlying mechanisms.”

Response: We thank the reviewer for the positive evaluation of our work. We hope that our answers here and changes in the revised manuscript address all the reviewer's concerns.

"My comments are fairly minor.

1) I wasn't sure if the regression analyses with the metrics was really necessary to report in the main text. As I understand it, these analysis basically demonstrate the the metrics are capturing a substantial amount of variance in the behavior. But I think this could just be mentioned in passing, since the main goal is to use the metrics not to explain behavior directly but to guide model development."

Response: We thank the reviewer for this suggestion. As the reviewer pointed out, much of the regression analyses originally included in the text were not directly relevant to the use of metrics to guide model development. However, another conclusion we were hoping to draw from these analyses was that different entropy-based metrics capture unique aspects of variance in deviation from matching. In the revised manuscript, we now have clarified this and moved most of the details of the regressions to a supplementary text. We now only include a description of the regression and R^2 values for each stepwise regression in the main text (see [R2.1] in the revised manuscript).

Accordingly, the first two regressions with behavioral metrics in the main text now read as follows:

(pages 8-9) *"To examine the relationship between existing behavioral metrics and undermatching, we first performed stepwise multiple regressions to predict deviation from matching for both mice and monkeys based on commonly used behavioral metrics including: $p(\text{win})$, $p(\text{stay})$, $p(\text{stay}|\text{win})$ and $p(\text{switch}|\text{lose})$. The threshold for adding a predictor was set at $p < 0.0001$ (see **Methods** for more details and **Supplementary Note 1** for regression equations). These regression models explained 31% and 34% of the variance in deviation from matching for mice and monkeys, respectively, which are significant but unsurprising amounts of overall variance (mice: Adjusted $R^2 = 0.31$; monkeys: Adjusted $R^2 = 0.34$).*

We next included the Repetition Index (RI) on the better (RI_B) and worse (RI_W) options (side or stimulus), which measure the tendency to stay beyond chance on the better and worse options, to predict undermatching (Soltani et al., 2013). To that end, we conducted additional stepwise multiple regressions that predicted deviation from matching using: RI_B , RI_W , $p(\text{win})$, $p(\text{stay})$, $p(\text{stay}|\text{win})$, and $p(\text{switch}|\text{lose})$ as predictors. These models explained 48% and 49% of the variance in deviation from matching for mice and monkeys, respectively (mice: Adjusted $R^2 = 0.48$; monkeys: Adjusted $R^2 = 0.49$). Thus, including RI_B and RI_W enabled us to account for additional 17% and 15% of variance, suggesting that staying beyond chance on both the better and worse choice options is a significant contributor to undermatching behavior [R2.1].”

The third regression with entropy-based metrics in the main text now reads as follows:

(pages 14-15) “To verify the utility of entropy-based metrics in predicting deviation from matching, we performed additional stepwise regressions to predict deviation from matching using our entropy-based metrics. In these models, we included $ERDS_+$, $ERDS_-$, $EODS_B$, $EODS_W$, $ERODS_{B+}$, $ERODS_{B-}$, $ERODS_{W+}$, $ERODS_{W-}$, RI_B , RI_W , $p(\text{win})$, $p(\text{stay})$, win-stay, and lose-switch as predictors (see **Supplementary Note 1** for regression equations).

These models explained 74% and 57% of total variance in deviation from matching for mice and monkeys, respectively (mice: Adjusted $R^2 = 0.74$; monkeys: Adjusted $R^2 = 0.57$). For mice, the regression model explained 26% more variance than the model with repetition indices and basic behavioral metrics and 43% more variance than the model with basic behavioral metrics only. For monkeys, the regression model explained 8% more variance than the model with repetition indices and basic behavioral metrics and 23% more variance than the model with basic behavioral metrics only. These are significant improvements over previous models, suggesting that most variance in undermatching behavior can be explained by trial-by-trial response to reward feedback.

In terms of the predictive power of different metrics, we found that for mice, the first three predictors added to the regression models were $ERODS_{W-}$ ($\Delta R^2 = 0.59$), $ERODS_{W+}$ ($\Delta R^2 = 0.04$), and $ERODS_{B+}$ ($\Delta R^2 = 0.02$). For monkeys, the first three predictors added were $ERODS_{W-}$ ($\Delta R^2 = 0.31$), $EODS_W$ ($\Delta R^2 = 0.09$), and $ERODS_{B+}$ ($\Delta R^2 = 0.06$). These results indicate that entropy-based metrics were the best predictors of deviation from matching when considering all metrics together. In addition, most entropy-based metrics included as predictors

were added to the final regression equations for both mice and monkeys. This suggests that despite their overlap, each entropy-based metric captures a unique aspect of the variance in deviation from matching behavior [R2.1].”

“2) Figure 4 is nearly unreadable. There’s just too much going on there. Maybe it could go in the supplement and be replaced with some condensed form that is more manageable.

Response: We thank the reviewer for pointing this out and for their helpful suggestion. We now have replaced original **Figure 4** it with the more readable Figure below that includes only Pearson correlations of different metrics with deviation from matching. This new figure still shows that entropy-based metrics are highly correlated with deviation from matching which was the primary purpose of including the previous **Figure 4** (that we now have included as supplemental **Figure S2**) (see [R2.2]).

Figure 3. Correlation between undermatching and proposed entropy-based metrics and underlying probabilities. (a) Pearson correlation between new and old behavioral metrics and deviation from matching in mice. Correlation coefficients are computed across all blocks, and metrics with non-significant correlations ($p > .0001$ to account for multiple comparisons) are indicated with a hollow bar. The metric with the highest correlation with deviation from matching is indicated with a star. **(b)** Similar to (a) but for monkeys. Overall, the new entropy-

based metrics show stronger correlation with deviation from matching than previous metrics [R2.2].

“3) In the Discussion, the authors address how negative choice weights can explain various phenomena in the decision making literature. I want to raise two issues here. First, there is more probability mass on positive choice weights in Fig S7i. Second, I think it’s somewhat problematic to assume that some phenomena are explained by positive choice weights and some by negative choice weights, consistent with many other past studies. Presumably this is not an idiosyncratic property of the subjects in these studies; possibly it could be a property of the tasks. However, this parameter is not task-dependent in the model. So the model doesn’t really offer a coherent explanation of why choice weights might be negative vs. positive. They are simply free parameters. I’m not arguing that the authors have to fill this gap in the current paper, just that they need to be somewhat careful in how they talk about what their model can or cannot explain.”

Response: We thank the reviewer for detailing these valid concerns. We agree with the reviewer that negative choice weights may not explain other decision-making phenomena as this could be task-dependent and moreover, the model does not explain why choice weights are positive or negative. We now have revised the corresponding section to reflect these two points and their relationship with previous literature (see [R2.3]).

Specifically, to address the first issue, we now state that we observe positive choice-memory effects in mice (more probability mass on positive choice weights) and the reverse in monkeys, and discuss the relationship of this finding to previous studies as follows:

(page 25) “We also observed weak, positive choice memory effects in mice such that mice tended to choose options that they had recently chosen. A previous study using a nearly identical task (reversal learning with same reward schedules (40/10) and baited rewards, but longer blocks) observed a much stronger, negative choice memory effect in mice (Fonseca et al., 2015). The reason for this difference is unclear given the similarity of the two tasks. Consistent with prior studies of choice history effects in monkeys (Lau and Glimcher, 2005), we identified strong, negative choice memory effects in monkeys such that the choice memory component encouraged switching from recently chosen options. Thus, the incorporation of the negative weights was only important for capturing behavior in the monkey task and therefore could be task dependent [R1.2] [R2.3].”

To address the second issue, we now have modified how we discuss the implications of our model for other decision-making phenomena in the literature to focus only on a similar task (Costa et. al 2015) and to clarify that the model can only replicate behavior, not explain it:

(pages 25-26) *“This negative weighting mechanism may be able to facilitate quick adaptation to reversals in monkeys, a behavior that has previously been suggested using a Bayesian approach (Costa et al., 2015), because negative weights in either the choice-memory or the loss-memory component encourage faster response to reversals. Future studies are needed to test whether this is the case [R2.3].”*

Finally, we also have included below text to the Discussion to point out the limitation of our approach:

(page 24) *“Although aforementioned differences in results for these two datasets may be partially explained by differences in task structure and species, they also highlight the limitations of using entropy-based metrics to guide model development. Entropy-based metrics describe properties of choice behavior that are helpful for making educated guesses about model structure, but alone, cannot provide a generative account of behavior [R1.5] [R2.3].”*

“4) This is very minor, but I don't see why you need to define the conditional entropy in terms of the mutual information in Eq. 7. You can just define it directly.”

Response: We thank the reviewer for pointing this out. We have modified equation 7 and removed equation 8 in the text to define conditional entropy directly (see [R2.4]). The more streamlined definition of conditional entropy now reads as follows:

(page 30) *“More specifically, ERDS is defined as the conditional entropy of using stay or switch strategy depending on win or lose in the preceding trial:*

$$ERDS = H(str|rew) = - \left(P(stay, win) \times \log_2 \left(\frac{P(stay, win)}{P(win)} \right) + P(switch, win) \times \log_2 \left(\frac{P(switch, win)}{P(win)} \right) + P(stay, lose) \times \log_2 \left(\frac{P(stay, lose)}{P(lose)} \right) + P(switch, lose) \times \log_2 \left(\frac{P(switch, lose)}{P(lose)} \right) \right) \quad (Eq. 7) [R2.4]”$$

REVIEWER COMMENTS

Reviewer #1 (Remarks to the Author):

The authors have responded to all my concerns and comments. Indeed, the response was refreshingly clear and straightforward, and I appreciate the amount of work that went into the revision. I have no further concerns and recommend publication in Nature Communications.

Reviewer #2 (Remarks to the Author):

I am satisfied with the response to my comments.